# REFRAG: RETHINKING RAG BASED DECODING

## ABSTRACT

Large Language Models (LLMs) have demonstrated remarkable capabilities in leveraging extensive external knowledge to enhance responses in multi-turn and agentic applications, such as retrieval-augmented generation (RAG). However, processing long-context inputs introduces significant system latency and demands substantial memory for the key-value cache, resulting in reduced throughput and a fundamental trade-off between knowledge enrichment and system efficiency. While minimizing latency for long-context inputs is a primary objective for LLMs, we contend that RAG systems require specialized consideration. In RAG, much of the LLM context consists of concatenated passages from retrieval, with only a small subset directly relevant to the query. These passages often exhibit low semantic similarity due to diversity or deduplication during re-ranking, leading to block-diagonal attention patterns that differ from those in standard LLM generation tasks. Based on this observation, we argue that most computations over the RAG context during decoding are unnecessary and can be eliminated with minimal impact on performance. To this end, we propose REFRAG, an efficient decoding framework that compresses, senses, and expands to improve latency in RAG applications. By exploiting this attention sparsity structure, we demonstrate a $30.85\times$ the time-to-first-token acceleration ($3.75\times$ improvement to previous work) without loss in perplexity. In addition, our optimization framework for large context enables REFRAG to extend the context size of LLMs by $16\times$. We provide rigorous validation of REFRAG across diverse long-context tasks, including RAG, multi-turn conversations, and long document summarization, spanning a wide range of datasets. Experimental results confirm that REFRAG delivers substantial speedup with no loss in accuracy compared to LLaMA models and other state-of-the-art baselines across various context sizes. Additionally, our experiments establish that the expanded context window of REFRAG further enhances accuracy for popular applications.

## 1 INTRODUCTION

Large Language Models (LLMs) have demonstrated impressive capabilities in contextual learning, leveraging information from their input to achieve superior performance across a range of downstream applications. For instance, in multi-turn conversations (Roller et al., 2021; Zhang et al., 2020), incorporating historical dialogue into the context enables LLMs to respond more effectively to user queries. In retrieval-augmented generation (RAG) (Guu et al., 2020; Izacard et al., 2022), LLMs generate more accurate answers by utilizing relevant search results retrieved from external sources. These examples highlight the power of LLMs to learn from context. However, it is well established that increasing prompt length for contextual learning leads to higher latency and greater memory consumption during inference (Yen et al., 2024). Specifically, longer prompts require additional memory for the key-value (KV) cache, which scales linearly with prompt length. Moreover, the time-to-first-token (TTFT) latency increases quadratically, while the time-to-iterative-token (TTIT) latency grows linearly with prompt length (Liu et al., 2025). As a result, LLM inference throughput degrades with larger contexts, limiting their applicability in scenarios demanding high throughput and low latency, such as web-scale discovery. Therefore, developing novel model architectures that optimize memory usage and inference latency is crucial for enhancing the practicality of contextual learning in these applications.

Optimizing inference latency for LLMs with extensive context is an active area of research, with approaches ranging from modifying the attention mechanism's complexity (Beltagy et al., 2020) to

sparsifying attention and context (Child et al., 2019; Xiao et al., 2024; Jiang et al., 2024), and altering context feeding strategies (Yen et al., 2024). However, most existing methods target generic LLM tasks with long context and are largely orthogonal to our work. This paper focuses on RAG-based applications, such as web-scale search, with the goal of improving inference latency, specifically, the TTFT. We argue that specialized techniques exploiting the unique structure and sparsity inherent in RAG contexts can substantially reduce memory and computational overhead. Treating RAG TTFT as a generic LLM inference problem overlooks several key aspects: 1) **Inefficient Token Allocation.** RAG contexts often contain sparse information, with many retrieved passages being uninformative and reused across multiple inferences. Allocating memory/computation for all the tokens, as we show in this paper, is unnecessarily wasteful. 2) **Wasteful Encoding and Other Information.** The retrieval process in RAG has already pre-processed the chunks of the contexts, and their encodings and other correlations with the query are already available due to the use of vectorizations and re-rankings. This information is discarded during decoding. 3) **Unusually Structured and Sparse Attention.** Due to diversity and other operations such as deduplication, most context chunks during decoding are unrelated, resulting in predominantly zero cross-attention between chunks (see Fig. 7).

### 1.1 OUR CONTRIBUTIONS

We propose REFRAG (REpresentation For RAG), a novel mechanism for efficient decoding of contexts in RAG. REFRAG significantly reduces latency, TTFT, and memory usage during decoding, all **without requiring modifications** to the LLM architecture or introducing new decoder parameters.

REFRAG makes several novel modifications to the decoding process: Instead of using tokens from retrieved passages as input, REFRAG leverages compressed chunk embeddings (optionally pre-computed) as approximate representations, feeding these embeddings directly into the decoder. This approach offers three main advantages: 1) It shortens the decoder's input length, improving token allocation efficiency; 2) It enables reuse of pre-computed chunk embeddings from retrieval, eliminating redundant computation; and 3) It reduces attention computation complexity, which now scales quadratically with the number of chunks rather than the number of tokens in the context.

Unlike prior methods (Yen et al., 2024), REFRAG supports compression of token chunks at arbitrary positions (see Fig. 1) while preserving the autoregressive nature of the decoder, thereby supporting multi-turn and agentic applications. The REFRAG decoder is continually pretrained to accept a mixed sequence of standard token embeddings and compressed chunk embeddings at arbitrary positions, allowing RAG context to be compressed at any turn and anywhere within the input. This "compress anywhere" capability is further enhanced by a lightweight reinforcement learning (RL) policy that selectively determines when full chunk token input is necessary and when low-cost, approximate chunk embeddings suffice . As a result, REFRAG minimizes reliance on computationally intensive token embeddings, condensing most chunks for the query in RAG settings.

We provide rigorous experimental validations of the effectiveness of REFRAG in continual pre-training and many real word long-context applications including RAG, multi-turn conversation with RAG and long document summarization. Results show that we achieve $30.75\times$ TTFT acceleration without loss in perplexity which is $3.75\times$ than previous method. Moreover, with extended context due to our compression, REFRAG achieves better performance than LLaMA without incurring higher latency in the downstream applications.

## 2 MODEL ARCHITECTURE

We denote the decoder model as $\mathcal{M}_{\text{dec}}$ and the encoder model as $\mathcal{M}_{\text{enc}}$. Given an input with $T$ tokens $x_1, x_2, \ldots, x_T$, we assume that the first $q$ tokens are main input tokens (e.g., questions) and the last $s$ tokens are context tokens (e.g., retrieved passages in RAG). We have $q + s = T$. For clarity, we focus on a single turn of question and retrieval in this section.

**Model overview.** Figure 1 shows the main architecture of REFRAG. This model consists of a decoder-only foundation model (e.g., LLaMA (Touvron et al., 2023)) and a lightweight encoder model (e.g., Roberta (Liu et al., 2019)). When given a question $x_1, \ldots, x_q$ and context $x_{q+1}, \ldots, x_T$ and , the context is chunked into $L := \frac{s}{k}$ number of $k$-sized chunks $\{C_1, \ldots, C_L\}$ where $C_i = \{x_{q+k*i}, \ldots, x_{q+k*i+k-1}\}$. The encoder model then processes all the chunks to obtain a chunk embedding for each chunk $\mathbf{c}_i = \mathcal{M}_{\text{enc}}(C_i)$. This chunk embedding is then projected with a projection layer $\phi$ to match the size of the token embedding of the decoder model, $\mathbf{e}_i^{\text{cnk}} = \phi(\mathbf{c}_i)$.

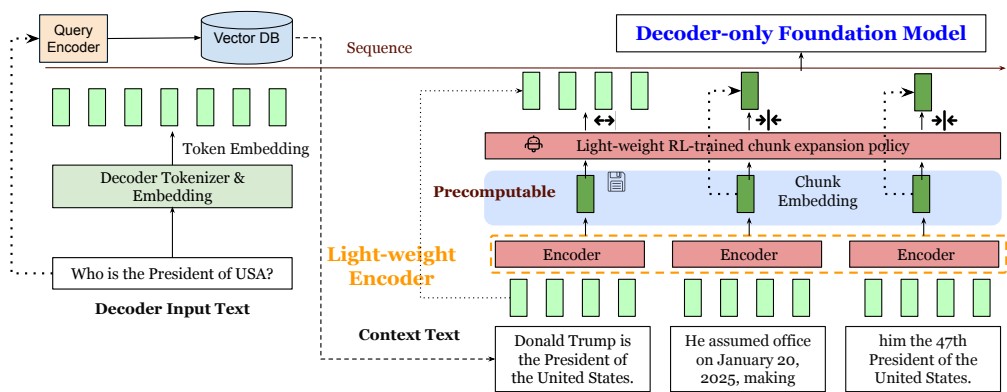

Figure 1: The main design of REFRAG. The input context is chunked and processed by the light-weight encoder to produce chunk embeddings, which are precomputable for efficient reuse. A light-weight RL policy decide few chunks to expand. These chunk embeddings along with the token embeddings of the question input are fed to the decoder.

These projected chunk embeddings are then fed to the decoder model along with the token embeddings for the question to generate the answer $y \sim \mathcal{M}_{\text{dec}}(\{\mathbf{e}_1, \ldots, \mathbf{e}_q, \mathbf{e}_1^{\text{cnk}}, \ldots, \mathbf{e}_L^{\text{cnk}}\})$ where $\mathbf{e}_i$ is the token embedding for token $x_i$. In real applications (e.g., RAG), the context is the dominating part of the input (i.e., $s \gg q$) and hence the overall input to the decoder will be reduced by a factor of $\simeq k$. This architectural design leads to significant reductions in both latency and memory usage, primarily due to the shortened input sequence. Additionally, an RL policy is trained to do selective compression to further improve the performance which we will defer the discussion to Section 2. Although REFRAG uses RL to select which compressed chunks to re-expand, the RL component is intentionally kept extremely lightweight. he action space is low-cardinality (a binary select/not-select decision per chunk), and training converges reliably using a stable negative next-paragraph perplexity reward. Next, we analyze the system performance gains achieved with a compression rate of $k$.

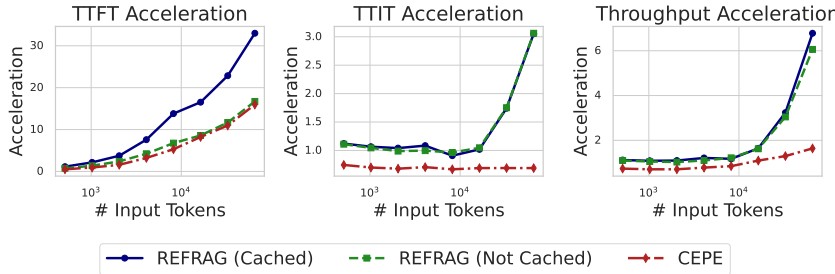

Figure 2: Empirical verification of inference acceleration of REFRAG with $k = 16$.

**Latency and throughput improvement.** We evaluate three metrics: TTFT, the latency to generate the first token; TTIT, the time to generate each subsequent token; and Throughput, the number of tokens generated per unit time. Theoretical analysis (Section B) shows that for short context lengths, our method achieves up to $k\times$ acceleration in TTFT and throughput. For longer context length, acceleration reaches up to $k^2\times$ for both metrics. Empirically, as shown in Fig. 2, with a context length of $16384$ (mid-to-long context), REFRAG with $k = 16$ achieves $16.53\times$ TTFT acceleration with cache and $8.59\times$ without cache[1], both surpassing CEPE ($2.01\times$ and $1.04\times$, respectively), while achieving 9.3% performance (measured by log-perplexity) compared to CEPE (Table 1). We achieve up to $6.78\times$ throughput acceleration compared to LLaMA, significantly outperforming CEPE. With

---

[1] REFRAG without cache means that we recompute the chunk embedding for the context and take this latency into account.

$k = 32$, TTFT acceleration reaches $32.99\times$ compared to LLaMA ($3.75\times$ compared to CEPE) while maintaining similar performance to CEPE (see Fig. 8 and Table 2). More detailed discussion on empirical evaluation is in Section B.

## 3 METHODOLOGY

To align the encoder and decoder, we follow the work of Yen et al. (2024) to use the next paragraph prediction tasks for continual pre-training (CPT). Specifically, for each data data point, it contains $s + o = T$ number of tokens, which we use for CPT to prepare the model for downstream tasks utilizing chunk embeddings. To further enhance performance, we introduce selective compression via RL. After aligning the encoder and decoder through CPT, we apply supervised fine-tuning (SFT) to adapt the model to specific downstream tasks, such as RAG and multi-turn conversation. Additional details are provided in Section 5.

During CPT, we input the first $s$ context tokens $x_{1:s}$ into the encoder and use its output to assist the decoder in predicting the next target $o$ tokens $x_{s+1:s+o}$. The encoder and decoder are processed in sequence. We can view the encoder to replace the embedding layer of the decoder. This task encourages the model to leverage contextual information for next-paragraph prediction, thereby equipping it for downstream applications. The objective is to align any encoder–decoder combination so that the generations produced with **compressed context** closely resemble those generated by the original decoder with access to the full context.

### 3.1 CONTINUAL PRE-TRAINING RECIPE

To ensure the success of the CPT phase, we propose a training recipe that incorporates a reconstruction task and a curriculum learning approach. Ablation studies in Section 4 demonstrate that **this recipe** is **crucial** for achieving strong CPT performance.

**Reconstruction task.** We input the first $s$ tokens $x_{1:s}$ to the encoder and learn to reconstruct tokens $x_{1:s}$ in the decoder. In this task, we **freeze the decoder model and only train the encoder and projection layer**. The main objectives are to align the encoder and projection layer so that: 1) encoder can compress $k$ tokens with minimal information loss, and 2) projection layer can effectively map the encoder's chunk embeddings into the decoder's token space, allowing the decoder to interpret and accurately reconstruct the original information. The reconstruction task was specifically chosen to encourage the model to rely on context memory rather than its parametric memory during training. Once the encoder is aligned with the decoder through this reconstruction task, we initiate CPT by **unfreezing the decoder**.

**Curriculum learning.** The training tasks described in the previous section may seem straightforward, but they are inherently complex. As the chunk length $k$ increases, the number of possible token combinations expands exponentially, specifically at a rate of $V^k$, where $V$ is the vocabulary size. Effectively capturing this diversity within a fixed-length embedding presents a significant challenge. Additionally, reconstructing $s = k \times L$ tokens from $L$ chunk embeddings further compounds the difficulty of the task.

*Counterintuitively, directly continuing pre-training of the decoder to utilize encoder outputs did not reduce perplexity, even for the reconstruction task.* To address the optimization challenge, we propose employing curriculum learning for both tasks. Curriculum learning incrementally increases task difficulty, enabling the model to gradually and effectively acquire complex skills. For the reconstruction task, training begins with reconstructing a single chunk: the encoder receives one chunk embedding $\mathbf{c}_1$ for $x_{1:k}$ and and the decoder reconstructs the $k$ tokens using the projected chunk embedding $\mathbf{e}_1^{cnk}$. Subsequently, the model reconstructs $x_{1:2k}$ from $\mathbf{e}_1^{cnk}, \mathbf{e}_2^{cnk}$, and so forth. To continuously adjust task difficulty, we vary the data mixture over time, starting with examples dominated by easier tasks (e.g., single chunk embedding) and gradually shifting towards those dominated by more difficult tasks (i.e., $L$ chunk embeddings). A visualization of the data mixture during curriculum learning is provided in Fig. 6, and the detailed scheduling is presented in Table 8.

**Selective compression** REFRAG introduces selective token compression, expanding important context chunks uncompressed to improve answer prediction. A RL policy, guided by next-paragraph prediction perplexity as a negative reward, a stable and low-variance reward, determines which chunks to retain in their original form. The encoder and decoder are fine-tuned to handle mixed inputs of compressed and uncompressed chunks. The policy network leverages chunk embeddings

and masking to optimize sequential chunk expansion, thereby preserving the decoder's autoregressive property and enabling flexible placement of compression. Further discussion on sequential selection is provided in Section B.1.

## 4 EXPERIMENTAL RESULTS

**Training datasets.** We use the Slimpajama dataset (Soboleva et al., 2023), an open source dataset for LLM pre-training. This dataset contains data from Wikipedia, Arxiv, Books, StackExchange, GitHub, Commoncrawl, C4. We only use the Book and ArXiv domains from the dataset since these two domains contain long texts (Yen et al., 2024). We sampled from this dataset to construct a 20B token training dataset which contains 50% data from Arxiv and 50% data from Book.

**Evaluation datasets.** We report the performance on the Book and ArXiv domain from Slimpajama which are hold out for evaluation only. To inspect the generalization of the model, we also report results on the PG19 (Rae et al., 2019) and Proof-pile datasets (Azerbayev et al., 2023).

**Baselines.** All baseline models are based on LLaMA-2-7B (Touvron et al., 2023), unless otherwise specified, to ensure fair comparison with prior work (Yen et al., 2024; Shi et al., 2024). Each data point contains $T = 4096$ tokens, split into $s = 2048$ context and $o = 2048$ output tokens. We evaluate perplexity on $x_{s+1:s+o}$. Below, we briefly describe the main baselines; further details are provided in Section C. **LLAMA-NO CONTEXT**: LLaMA-2-7B evaluated on $x_{s+1:s+o}$ with only output tokens as input. **LLAMA-FULL CONTEXT**: LLaMA-2-7B evaluated on $x_{s+1:s+o}$ with the full sequence $x_{1:T}$ as input. **CEPE**: Memory-efficient long-context model (Yen et al., 2024) a previous SOTA model which share some similarity to REFRAG CEPED denotes its instruction-tuned variant. **LLAMA-32K**: LLaMA-2-7B fine-tuned for 32K context length. **RE-PLUG**: Retrieval-augmented LLaMA-2-7B (Shi et al., 2024). **REFRAG**: Our approach (see Figure 1); $\text{REFRAG}_k$ denotes compression rate $k$, $\text{REFRAG}_{\text{RL}}$ uses RL-based selective compression. **LLAMA**$_K$: LLaMA-2-7B evaluated on $x_{s+1:s+o}$ with the truncated sequence $x_{s-K:T}$ as input to match the token count of REFRAG.

Table 1 reports performance for $s = 2048$ and $o \in \{512, 1024, 2048\}$, where, e.g., P512 denotes $o = 512$. Bolded results compare baselines, excluding LLAMA-FULL CONTEXT and LLAMA-32K, which use full context without compression and are expected to perform best. Notably, $\text{REFRAG}_8$ and $\text{REFRAG}_{16}$ consistently outperform other baselines across nearly all settings, while also achieving lower latency than CEPE (Fig. 2). For reference, $\text{LLAMA}_{256}$ uses only the last 256 tokens, matching the number of chunk embeddings in $\text{REFRAG}_8$ ($s/k = 256$), yet $\text{REFRAG}_8$ consistently surpasses $\text{LLAMA}_{256}$, demonstrating the effectiveness of compressed chunk embeddings.

Table 2 evaluates $o = 2048$ with extended context lengths $s \in \{4096, 8192, 16384\}$. Although our model is trained on $s + o = 6144$, both $\text{REFRAG}_8$ and $\text{REFRAG}_{16}$ maintain superior performance at longer contexts. The original Llama-2-7B supports only a 4k context window, whereas our approach enables extrapolation via chunk embeddings, extending context and supporting broader applications.

*With a compression rate of 16, we achieve a 9.3% average log-perplexity improvement over CEPE across four datasets[2]. Meanwhile, our method is 16.53× faster than LLaMA in TTFT and 2.01× faster than CEPE (Section C.4). At a compression rate of 32, our log-perplexity matches CEPE, while TTFT acceleration increases to 30.85× over LLaMA and 3.75× over CEPE.*

Figure 3 presents the performance of various methods for selective compression. We expand $p$ fraction of the chunks in the original token space using the RL policy. The effective compression rate $\frac{k}{1-p+kp}$ decreases when fewer chunks are compressed (i.e., $p$ increases). We compare the perplexity of $x_{s+1:s+o}$ using different selection policy under different $p$. The perplexity-based selection is an heuristic based selection which compresses chunks with low perplexity (Perplexity-desc) or high perplexity (Perplexity-asc). The perplexity is measured by the LLaMA-2-7B model. Intuitively, a chunk with lower perplexity contains less information and can therefore be compressed with minimal information loss. Ideally, this approach should outperform random selection, which is indeed observed in Fig. 3. The RL-based selective compression policy consistently achieves superior performance across varying compression rates $p$.

---

[2]Percentage calculated as $\frac{\text{LLAMA-NO CONTEXT} - \text{Log-perplexity to inspect}}{\text{LLAMA-NO CONTEXT} - \min(\text{LLAMA-FULL CONTEXT}, \text{LLAMA-32K})}$

Table 1: Log-Perplexity on output tokens $x_{s+1:s+o}$ given context tokens $x_{1:s}$ for different models. We use $s = 2048$ and $o \in \{512, 1024, 2048\}$ here. Bolding are based on comparing baselines excluding LLAMA-FULL CONTEXT and LLAMA-32K since they are expected to be the best (ideally). The lower the better ($\downarrow$).

| | Arxiv | | | Book | | | PG19 | | | ProofPile | | |
|---|---|---|---|---|---|---|---|---|---|---|---|---|
| | P512 | P1024 | P2048 | P512 | P1024 | P2048 | P512 | P1024 | P2048 | P512 | P1024 | P2048 ↓ |
| LLAMA-FULL CONTEXT | 1.075 | 1.074 | 1.069 | 1.830 | 1.827 | 1.826 | 1.947 | 1.941 | 1.935 | 0.952 | 0.940 | 0.931 |
| LLAMA-32K | 1.086 | 1.084 | 1.076 | 1.887 | 1.883 | 1.880 | 1.988 | 1.982 | 1.975 | 0.961 | 0.948 | 0.937 |
| LLAMA-NO CONTEXT | 1.526 | 1.371 | 1.254 | 2.101 | 1.995 | 1.927 | 2.211 | 2.102 | 2.030 | 1.437 | 1.256 | 1.127 |
| LLAMA$_{256}$ | 1.267 | 1.221 | 1.171 | **1.924** | 1.897 | 1.874 | 2.031 | 2.003 | 1.978 | 1.156 | 1.094 | 1.038 |
| REPLUG | 1.526 | 1.371 | 1.254 | 2.101 | 1.995 | 1.927 | 2.211 | 2.102 | 2.030 | 1.437 | 1.256 | 1.127 |
| CEPE | 1.196 | 1.148 | 1.107 | 1.946 | 1.896 | 1.864 | 2.057 | 2.002 | 1.964 | 1.075 | 1.014 | 0.968 |
| REFRAG$_8$ | **1.124** | **1.091** | **1.062** | **1.905** | **1.868** | **1.844** | **1.996** | **1.956** | **1.927** | **0.997** | **0.952** | **0.916** |
| REFRAG$_{16}$ | **1.157** | **1.114** | **1.076** | 1.925 | **1.882** | **1.853** | **2.016** | **1.971** | **1.938** | **1.034** | **0.976** | **0.931** |
| REFRAG$_{32}$ | 1.215 | 1.154 | **1.103** | 1.946 | **1.896** | **1.862** | 2.039 | **1.987** | **1.949** | 1.097 | 1.020 | **0.961** |

Table 2: Log-Perplexity on output tokens $x_{s+1:s+o}$ given different length of context. We use $s \in \{4096, 8192, 16384\}$ and $o = 2048$ here. Bolding are based on comparing baselines excluding LLAMA-FULL CONTEXT and LLAMA-32K since they are expected to be the best (ideally). The lower the better ($\downarrow$).

| | Context Length =4096 | | | | Context Length=8192 | | | | Context Length=16384 | | | |
|---|---|---|---|---|---|---|---|---|---|---|---|---|
| | Arxiv | Book | PG19 | ProofPile | Arxiv | Book | PG19 | ProofPile | Arxiv | Book | PG19 | ProofPile ↓ |
| LLAMA-FULL CONTEXT | 6.751 | 6.956 | 6.829 | 6.701 | 9.675 | 9.069 | 8.963 | 9.401 | 9.043 | 8.913 | 8.848 | 8.989 |
| LLAMA-32K | 1.037 | 1.862 | 1.960 | 0.898 | 0.965 | 1.867 | 1.947 | 0.834 | 0.865 | 1.840 | 1.943 | 0.770 |
| LLAMA-NO CONTEXT | 1.253 | 1.925 | 2.030 | 1.126 | 1.226 | 1.949 | 2.032 | 1.110 | 1.174 | 1.939 | 2.041 | 1.081 |
| REPLUG | 1.253 | 1.925 | 2.030 | 1.126 | 1.226 | 1.949 | 2.032 | 1.110 | 1.174 | 1.939 | 2.041 | 1.081 |
| CEPE | 1.085 | 1.856 | 1.959 | 0.945 | 1.032 | 1.878 | 1.958 | 0.904 | 0.960 | 1.864 | 1.966 | 0.863 |
| REFRAG$_8$ | **1.042** | **1.837** | **1.922** | **0.894** | **0.983** | **1.839** | **1.922** | **0.858** | 0.977 | **1.840** | **1.939** | 0.891 |
| REFRAG$_{16}$ | **1.058** | **1.847** | **1.934** | **0.910** | **0.994** | **1.845** | **1.932** | **0.871** | **0.942** | **1.840** | **1.945** | **0.850** |
| REFRAG$_{32}$ | **1.088** | **1.857** | **1.946** | **0.944** | **1.032** | **1.860** | **1.945** | **0.912** | **0.969** | **1.852** | **1.955** | **0.880** |

Table 3: Log-Perplexity of continual pre-training for different encoder-decoder combinations. Lower log-perplexity indicates better performance.

| Encoder–Decoder | Context Length | LLaMA-3.1-8B | | | LLaMA-3.2-3B | | |
|---|---|---|---|---|---|---|---|
| | | P512 | P1024 | P2048 | P512 | P1024 | P2048 ↓ |
| Full Context | 2048 | 1.000 | 0.989 | 0.972 | 1.092 | 1.080 | 1.062 |
| No Context | 0 | 1.445 | 1.286 | 1.162 | 1.559 | 1.392 | 1.262 |
| Roberta-Base | 2048 | 1.109 | 1.067 | 1.026 | 1.175 | 1.133 | 1.093 |
| Roberta-Large | 2048 | 1.107 | 1.065 | 1.025 | 1.170 | 1.130 | 1.091 |
| Roberta-Base | 4096 | 1.067 | 1.032 | 0.999 | 1.142 | 1.105 | 1.070 |
| Roberta-Large | 4096 | 1.065 | 1.031 | 0.998 | 1.130 | 1.096 | 1.064 |

## 4.1 ABLATION STUDY

**Curriculum learning is essential for effective training in the reconstruction task.** The reconstruction task, while intuitive, is particularly challenging when multiple chunks must be reconstructed. Table 11 shows the performance of the reconstruction task with and without curriculum learning (i.e., reconstruction of $x_{1:s}$ from $s/k$ chunk embedding directly). The results indicate that curriculum learning is essential for the success of the reconstruction task.

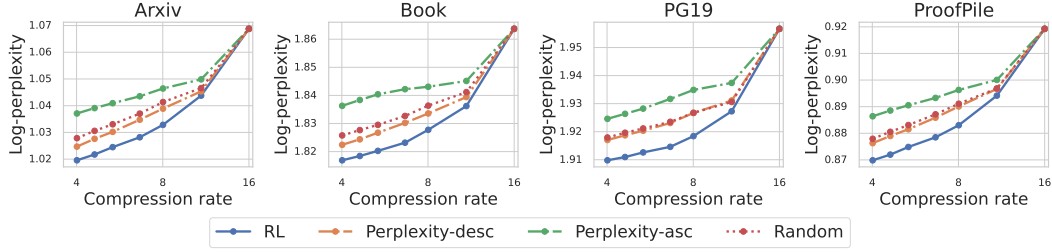

Figure 3: Log-Perplexity on $x_{s+1:s+o}$ under varying compression rates by selectively compressing different percentages of chunks. We compare three selection methods: **RL** (trained policy), **Perplexity-desc** (heuristic: lower perplexity), **Perplexity-asc** (heuristic: higher perplexity), and **Random** (random selection).

**Reconstruction task is essential for the model to learn the continual pre-training task.** Table 12 shows the performance of the continual pre-training task with and without initialization from the reconstruction task. The results indicate that pre-training on the reconstruction task is important for the success of continual pre-training.

**Advantages of RL-based selective compression.** Figure 3 under various compression rates, achieved by varying the number of chunks to compress (i.e., adjusting $p$). Notably, a compression rate of 8 can be obtained either by configuring $REFRAG_{16}$ to compress the appropriate number of chunks, or by employing $REFRAG_8$ with full compression, which is natively trained at a compression rate of 8. This raises a natural question: does the former approach outperform the latter? Table 13 demonstrates that $REFRAG_{16}$ with RL-based selective compression consistently outperforms $REFRAG_8$ across different datasets and context lengths. This finding is particularly surprising, as $REFRAG_{16}$ achieves a compression rate of 8 without recomputing chunk embeddings, yet still surpasses the performance of $REFRAG_8$. These results further highlight the effectiveness of the RL-trained policy and underscore the practicality of dynamically adjusting the compression rate without compromising performance.

**REFRAG trained under different compression rates.** Figure 10 illustrates the training trajectory of REFRAG under different compression rates in the continual pre-training task. We observe a performance regression as the compression rate increases; however, even at a compression rate of 32, our model remains competitive (as shown in Table 1). In contrast, a compression rate of 64 appears to be overly aggressive, resulting in diminished performance. These findings suggest a practical limit to the compression rate beyond which the model's capability is significantly reduced.

**Different combinations of encoder and decoder models for REFRAG.** We employ LLaMA-2-7B and LLaMA-2-13B as decoders, and RoBERTa-Base and RoBERTa-Large as encoders, to investigate how model performance varies with different encoder and decoder sizes. Figure 11 presents results for various encoder-decoder combinations. We observe that increasing the number of parameters in the decoder leads to a substantial reduction in loss, whereas enlarging the encoder yields only a modest improvement. This discrepancy may be attributed to the relatively minor increase in size from RoBERTa-Base to RoBERTa-Large compared to the substantial jump from 7B to 13B in the decoder. Additional results in Fig. 12 indicate that a larger encoder may not always be advantageous when training with limited data in the continual pre-training setting. This observation aligns with previous findings by Li et al. (2024), which demonstrate that larger encoders in multi-modal models can negatively impact performance when data is scarce. To further validate our training approach on other decoder models, we conduct experiments with LLaMA-3.1-8B and LLaMA-3.2-3B. Table 3 reports the performance of these models paired with RoBERTa-Base and RoBERTa-Large encoders on the Arxiv domain. Models trained with our recipe achieve performance comparable to the Full Context setting (i.e., without context compression). Importantly, we observe that REFRAG's pretraining behavior is consistent across model families across LLaMA 2 and LLaMA 3 family. Moreover, increasing the context length continues to benefit our model, as evidenced by lower perplexity for a context length of 4096 compared to 2048.

## 5 CONTEXTUAL LEARNING APPLICATIONS

In this section, we investigate fine-tuning the model obtained from the pre-training stage to address various downstream tasks, including RAG, long document summarization, and multi-turn conversation with RAG. These metric (accuracy-based, or ROUGE-based) complement the perplexity metric used in our continued pretraining experimental section. For each application, we curate an instruction-tuning dataset to facilitate model fine-tuning.

### 5.1 RETRIEVAL AUGMENTED GENERATION

**Training dataset.** We follow the work of Lin et al. (2024) and use a combination of question answering datasets from 5 domains to fine-tune our model, which contains 1.1 million data points.

**Dialogue**: OpenAssistant Conversations Dataset. **Open-Domain QA**: CommonsenseQA, MathQA, Web Questions, Wiki Question Answering, Yahoo! Answers QA, FreebaseQA, MS MARCO. **Reading Comprehension**: Discrete Reasoning Over Paragraphs, PubMedQA, QuaRel, SQuADv2. **Chain-of-thought Reasoning**: Algebra QA with Rationales, Explanations for CommonsenseQ, Grade School Math 8K, MathQA, StrategyQA.

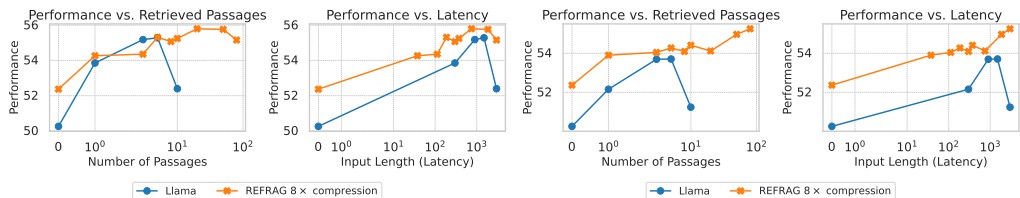

Figure 4: RAG performance comparison under a strong retriever scenario (left) and a weak retriever scenario and a strong retriever scenario (right). REFRAG perform similarly to LLaMA model under the same retrieved passages (slightly better in a weaker retriever case) while outperform significantly under the same latency.

**Evaluation dataset.** We hold out 5% of the data for each dataset in the training dataset for evaluation. Additionally, we use the datasets that are commonly used in RAG literature (Izacard et al., 2023b; Lin et al., 2024), including MMLU (Hendrycks et al., 2021), BoolQ (Clark et al., 2019), SIQA (Sap et al., 2019), PIQA (Bisk et al., 2020), and Knowledge Intensive Language Tasks (KILT) (Petroni et al., 2020) (including HellaSwag, Winogrande, TQA, FEVER, NQ). We evaluate our performance on 2 settings: 1) **Strong Retriever**: In this setting we use a strong retriever and retrieve the K-nearest neighbors to answer the question; 2) **Weak Retriever**: Real-world retrieval quality varies due to approximate vector search, aggressive deduping, and user queries lacking specificity. We simulate extreme retrieval noise using a random K-of-200 retriever. In this setting we retrieve 200 passages and pick random K passages to answer the question. The weak retriever setting closely resembles real-world systems, as RAG retrieval systems often suffer from error accumulation across subsystems.

**Baselines.** We compare REFRAG with LLAMAfinetuned on the same supervised finetuning dataset (SFT). We also add comparison with CEPED, and REPLUG on the same benchmarks. A table summarizing the evaluation metrics for each dataset is included in Table 7. We always use 0-shot prompting as this is natural settings for RAG. Note that during decoder generation, REFRAG never compresses model-generated outputs (or CoTs); only the retrieved context is compressed. However, the performance of CEPEDin our RAG evaluations differs from its original report primarily due to differences in instruction-tuning datasets and evaluation protocol. Our RAG setup uses strict zero-shot generation with concise factual answering, while the released CEPE checkpoint follows a different instruction style and produces longer outputs, leading to lower exact match (EM) under our protocol. This affects the downstream RAG evaluation. Importantly, CEPE and REFRAG use the same CPT dataset and the same LLaMA-7B architecture; thus the pretraining-level comparisons are fully comparable and aligned.

**Retriever and retrieval corpus.** We follow the work of Lin et al. (2024) to use Wikipedia dumps and CommonCrawl dumps to create a retrieval corpus with 400 million passages. Each passage contains less than 200 words. We use the DRAGON+ model Lin et al. (2023) as our retriever and use the implementation of Izacard et al. (2023a) to retrieve the K-nearest neighbors as the retrieved passages for each question. Note that we compute chunk embeddings dynamically using the lightweight encoder and they are not cached.

**Result analysis.** Table 4 shows the performance of different baselines under short and long contexts (i.e., varying number of retrieved passages)[3]. (1/# tokens) is inverse for the number of tokens in the decoder model. This is used as a metric to gauge the latency of the model (the higher, the lower latency). LLAMA$_{FT}$ is the original LLaMA-2-7B model that is fine-tuned on the same RAG dataset used to train our model. We compare the performance under both the short context and the long context scenarios. For the short context, we use 1 passage for LLAMA$_{FT}$ and use 8 passages for all our models. The baseline of REFRAG$_8$ will have the same latency as the LLAMA$_{FT}$ model. However, due to the compression, we are able to have more context information and hence achieve better performance. Surprisingly, REFRAG$_{16}$ and REFRAG$_{32}$ both outperform the LLAMA$_{FT}$ model despite having $2\times$ and $4\times$ fewer tokens in the decoder (i.e., lower latency). The same result occurs in long context scenarios. Our model has even higher performance gains in multi-choice

---

[3]Note that the implementation of our exact match is stricter than other works. We follow the work of Lin et al. (2024) to use the stricter version and hence the reported numbers are lower in general.

tasks. Table 15 shows the performance of our model under different numbers of passages. The result suggests that most tasks still benefit from more passages in our model. Figure 4 shows the performance averaged over all 16 tasks in Table 4 for both strong retriever and weak retriever setting. The result demonstrates that under the same number of retrieved passages, we are able to match the performance of LLaMA in the strong retriever setting and even outperform LLaMA under the weak retriever setting. This is because our model enables larger context and hence enables extract more useful information when the retrieved passages are less relevant. **Under equivalent latency constraints, REFRAG consistently outperform LLaMA on both settings as the saved context can be reinvested to include additional information within the same latency budget**.

Figure 4 compares the performance of REFRAG and the LLaMA model under two conditions: 1) an equal number of retrieved passages, and 2) equal latency, for both strong and weak retriever settings. *With a strong retriever and a maximum of 10 passages,* REFRAG *matches LLaMA's performance while achieving a* $5.26\times$ *speedup in TTFT. At equal latency (8 passages for* REFRAG *vs. 1 for LLaMA),* REFRAG *attains a* $1.22\%$ *average improvement across 16 RAG tasks. With a weak retriever setting, at 10 passages,* REFRAG *improves performance by* $0.71\%$ *and accelerates TTFT by* $5.26\times$ *compared to LLaMA. At equal latency (8 passages for* REFRAG *vs. 1 for LLaMA),* REFRAG *achieves a* $1.93\%$ *average gain over 16 RAG tasks.*

Table 4: Comparison of model performance of different models with different number of retrieved passages for RAG under the strong retriever scenario.

| Generation | NQ | FEVER | TQA | WebQA | FreebaseQA | GSM8K | StrategyQA | BoolQ ↑ | (1/ # tokens) |
|---|---|---|---|---|---|---|---|---|---|
| Short context with the same latency | | | | | | | | | |
| LLAMA$_{FT}$ + 1 passage | **23.96** | 62.04 | 9.64 | 37.33 | **75.18** | 7.38 | 64.44 | **29.24** | 1× |
| REFRAG$_8$+ 8 passages | 22.96 | 66.59 | **13.05** | 38.67 | 73.46 | 7.38 | 75.56 | 3.30 | 1× |
| REFRAG$_{16}$+ 8 passages | 22.94 | **62.88** | 12.97 | 42.67 | 71.50 | **9.40** | 71.11 | 5.87 | 2× |
| REFRAG$_{32}$+ 8 passages | 22.11 | **64.24** | 12.57 | **41.33** | 71.74 | **12.75** | **73.33** | 1.99 | 4× |
| Long context | | | | | | | | | |
| LLAMA$_{FT}$ + 10 passages | **26.08** | 65.44 | 9.68 | **40.00** | **76.17** | 6.71 | 68.89 | 30.00 | 1× |
| LLAMA-32K +80 passages | 1.24 | 0.14 | 0.52 | 10.67 | 9.83 | 0.00 | 0.00 | 0.03 | |
| REFRAG$_8$ +80 passages | 24.15 | 68.83 | **13.06** | 37.33 | 74.20 | 7.38 | 71.11 | 7.03 | 1× |
| REFRAG$_{16}$ +80 passages | 23.30 | 66.01 | 12.65 | 38.67 | 75.43 | **12.08** | 73.33 | 12.23 | 2× |
| REFRAG$_{32}$ +80 passages | 23.02 | 68.48 | 12.14 | 38.67 | 71.74 | 9.40 | 68.89 | 6.42 | 4× |
| CEPED* +80 passages | 0.03 | 65.68 | 0.01 | 0.00 | 0.00 | 0.00 | 0.00 | **57.80** | |
| REPLUG* +80 passages | - | - | - | - | - | - | 64.44 | - | |
| **Multi-Choice** | MMLU | CommonsenseQA | MathQA | ECQA | HellaSwag | SIQA | PIQA | Winogrande ↑ | |
| Short context with the same latency | | | | | | | | | |
| LLAMA$_{FT}$ + 1 context | **50.23** | 85.05 | 99.50 | 84.77 | 41.80 | 68.12 | 67.36 | 55.64 | 1× |
| REFRAG$_8$ + 8 passages | 50.29 | 92.27 | **99.66** | **94.70** | **45.23** | **68.94** | **71.38** | **57.70** | 1× |
| REFRAG$_{16}$ + 8 passages | 49.84 | **89.18** | **99.66** | **98.01** | 39.33 | 68.42 | 70.29 | 56.67 | 2× |
| REFRAG$_{32}$ + 8 passages | 49.51 | **91.75** | **99.50** | **97.35** | **42.86** | 68.17 | 68.34 | 56.75 | 4× |
| Long context | | | | | | | | | |
| LLAMA$_{FT}$ + 10 passages | 48.66 | 82.99 | 68.46 | 84.11 | 41.77 | 67.45 | 68.01 | 53.91 | 1× |
| LLAMA-32K +80 passages | 22.21 | 16.49 | 19.80 | 16.56 | 23.76 | 24.16 | 34.17 | 48.86 | |
| REFRAG$_8$ +80 passages | **50.42** | 92.27 | **99.66** | 97.35 | **44.61** | 68.22 | 69.37 | **57.54** | 1× |
| REFRAG$_{16}$ +80 passages | **50.88** | 89.69 | **99.66** | 96.69 | 38.50 | **68.47** | 70.89 | 56.99 | 2× |
| REFRAG$_{32}$ +80 passages | 49.77 | 90.72 | **99.50** | **98.01** | 43.37 | **68.47** | 69.04 | 56.99 | 4× |
| CEPED* +80 passages | 26.26 | 26.29 | 23.66 | 24.50 | 24.95 | 32.86 | 48.53 | 44.51 | |
| REPLUG* +80 passages | - | 78.35 | - | 76.16 | - | 65.51 | - | - | |

- means the corresponding model has out-of-memory error.
* For CEPED, REPLUG, we reuse the instruction-tuned CEPE checkpoint which is finetuned on different SFT datasets. See Section 5.1 for details.

## 5.2 MULTI-TURN CONVERSATION

We use three different knowledge-intensive multi-turn conversation datasets: TopiOCQA (Adlakha et al., 2022), ORConvQA (Qu et al., 2020), and QReCC (Anantha et al., 2021). For each conversation turn, we retrieve $K$ passages using the same retriever and retrieval corpus as described in Section 5.1.

**Result analysis.** Table 5 presents results across varying numbers of conversational turns and retrieved passages. Our model outperforms LLAMA$_{FT}$ on two out of three datasets in the 5-passage setting, and on all three datasets in the 10-passage setting. This improvement is attributable to the limited 4k-token context window of LLAMA$_{FT}$, which necessitates truncating portions of the conversational history in longer contexts, resulting in the loss of crucial information required to answer subsequent questions. In contrast, our model, trained on the same LLAMA model without extending its effective positional encoding, maintains robust performance even with a large number of passages, owing to the benefits of our compression approach. Table 14 further reports the performance of different models under varying numbers of passages, with our model consistently achieving superior results on two out of three datasets for the reasons outlined above.

Table 5: Performance on multi-turn RAG tasks for # Passages = 5 and # Passages = 10.

| | # Turns ($\geq$) | ORConvQA | QReCC | TopiOCQA ↑ | | # Turns ($\geq$) | ORConvQA | QReCC | TopiOCQA ↑ |
|---|---|---|---|---|---|---|---|---|---|
| | | # Passages = 5 | | | | | # Passages = 10 | | |
| LLAMA$_{FT}$ | 2 | 20.73 | **18.72** | 26.98 | LLAMA$_{FT}$ | 2 | 16.52 | 17.31 | 23.02 |
| REFRAG$_8$ | 2 | **21.17** | 17.73 | 28.04 | REFRAG$_8$ | 2 | **21.15** | **17.92** | 27.97 |
| REFRAG$_{16}$ | 2 | 20.19 | 17.30 | 27.89 | REFRAG$_{16}$ | 2 | 20.79 | 17.37 | **28.45** |
| REFRAG$_{32}$ | 2 | 19.70 | 17.35 | **28.67** | REFRAG$_{32}$ | 2 | 19.67 | 17.16 | 28.31 |
| LLAMA$_{FT}$ | 4 | 20.33 | **16.42** | 23.50 | LLAMA$_{FT}$ | 4 | 16.90 | 14.69 | 20.23 |
| REFRAG$_8$ | 4 | 22.78 | 15.61 | 26.93 | REFRAG$_8$ | 4 | **22.63** | **15.68** | 25.95 |
| REFRAG$_{16}$ | 4 | **21.94** | 15.27 | **27.03** | REFRAG$_{16}$ | 4 | 21.84 | 15.21 | **26.12** |
| REFRAG$_{32}$ | 4 | 21.68 | 15.45 | 26.45 | REFRAG$_{32}$ | 4 | 21.75 | 15.33 | 25.77 |
| LLAMA$_{FT}$ | 6 | 20.76 | **11.92** | 23.10 | LLAMA$_{FT}$ | 6 | 14.44 | 10.72 | 19.52 |
| REFRAG$_8$ | 6 | **23.11** | 10.88 | 25.37 | REFRAG$_8$ | 6 | 20.59 | **11.00** | 25.16 |
| REFRAG$_{16}$ | 6 | 21.69 | 10.75 | **26.17** | REFRAG$_{16}$ | 6 | 21.05 | 10.50 | 24.96 |
| REFRAG$_{32}$ | 6 | 21.19 | 10.69 | 25.51 | REFRAG$_{32}$ | 6 | **21.67** | 10.79 | **25.23** |

## 5.3 SUMMARIZATION TASK

We fine-tune our model on the long document summarization dataset (Cohan et al., 2018). This dataset contains long scientific articles from Arxiv and Pubmed, and the task is to generate the abstract given the entire article. This application is challenging due to the long-context nature of the task. We fine-tune the REFRAG and LLAMA models on these two datasets and report the performance on the validation set. The summarization task provides an ideal condition to inspect whether it is beneficial to bring more information with compressed representation or less information without compression, since correct summarization requires complete information from the whole document.

**Result analysis.** Table 21 shows the performance of different baselines under the same number of tokens in the decoder. REPLUG$_{FT}$ means that we adopt the REPLUG framework using LLAMA$_{FT}$, and REPLUG$_{Chat}$ means that we adopt the LLaMA-2-7B-Chat model for REPLUG. We did not report some of our methods for certain decoder token counts since there were not enough input tokens for those compression rates. Our model achieves the best performance under the same number of decoder tokens (i.e., same latency). Additionally, REFRAG$_{16}$ performs better than REFRAG$_8$ at a decoder token count of 128, since the former model is able to incorporate more information from the document with a higher compression rate.

## 5.4 DISCUSSION ON QUALITY-LATENCY TRADE-OFFS

We note that REFRAG performs lossy compression of the input context; thus some degradation in performance (e.g., perplexity, or automatic metrics) under a fixed raw context with maximal latency gain is expected. In the experimental section, we address the key question of **whether the resulting trade-off is favorable** for REFRAG. For the summarization task, (Table 21), we observe that under the same raw tokens (1028 LLAMA tokens vs 128 REFRAG-8 tokens), compressing context yields a $\sim 12\%$ ROUGE-1 relative drop for REFRAG on the Arxiv dataset. However, when comparing under the same effective context (i.e., fixed latency budget), REFRAG outperforms LLAMA. Similar patterns are observed in the Pubmed dataset and RAG evaluations (Table 4, Fig. 4). Under same latency budget, REFRAG consistently outperforms LLAMA because it leverages compression to process more content. Moreover, the RL-based selective expansion policy further improves the quality–latency trade-off for REFRAG by expanding to full tokens only when necessary, yielding consistently improved trade-offs (Table 13). This exposes a **quality-latency frontier** that conventional decoding cannot achieve.

## 6 CONCLUSION

In this work, we introduced REFRAG, a novel and efficient decoding framework tailored for RAG applications. By leveraging the inherent sparsity and block-diagonal attention patterns present in RAG contexts, REFRAG compresses, senses, and expands context representations to significantly reduce both memory usage and inference latency, particularly the TTFT. Extensive experiments across a range of long-context applications, including RAG, multi-turn conversations, and long document summarization, demonstrate that REFRAG achieves up to $30.85\times$ TTFT acceleration ($3.75\times$ over previous state-of-the-art methods) without any loss in perplexity or downstream accuracy. Our results highlight the importance of specialized treatment for RAG-based systems and open new directions for efficient large-context LLM inference. We believe that REFRAG provides a practical and scalable solution for deploying LLMs in latency-sensitive, knowledge-intensive applications.

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

# A    RELATED WORKS

**Retrieval-Augmented Language Modeling.**    Recent research has extensively investigated novel model architectures to improve retrieval-augmented generation. Guu et al. (2020) introduced pre-training for retrieval-augmented masked language models. Building on this, Borgeaud et al. (2022) proposed a new architecture and pre-training paradigm for generative LLMs, leveraging cross-attention and end-to-end pre-training with retrieval from a trillion-token data store, achieving strong performance. Subsequent work by Shi et al. (2024) and Lin et al. (2024) focused on fine-tuning existing LLMs by prepending retrieved passages to prompts and employing ensemble methods for response generation. Additionally, Izacard & Grave (2021) introduced fusion-in-decoder, which uses an encoder to process each passage in parallel and concatenates the hidden states for generation via a decoder. This approach accelerates attention computation by removing cross-document attention, but does not apply compression in the decoder, which could further reduce latency.

**Efficient Long-Context LLMs.**    Recent research has investigated various strategies to reduce memory usage and accelerate latency in long-context generation for LLMs. Choromanski et al. (2021) introduced compressed attention, reducing attention complexity from quadratic to linear; however, this method does not address memory requirements. It is complementary to our approach and can be integrated to further improve latency. StreamingLLM(Xiao et al., 2024) proposed attention sinks to decrease KV cache memory for long-context generation, though this does not reduce latency during the pre-filling stage. CEPE (Yen et al., 2024) employs cross-attention to token embeddings from context tokens, reducing both KV cache memory and attention computations. However, CEPE is limited to prefix context applications, as it disrupts the causal structure of the context, making it unsuitable for tasks such as multi-turn RAG or summarization. Additionally, CEPE does not utilize token compression, resulting in similar or even increased decoding latency. Concurrently with our work, Dai et al. (2025) proposed PCC, an embedding-based memory mechanism that summarizes past context into compact vectors, enabling retrieval of salient information during subsequent processing. Like CEPE, PCC is limited to prefix context applications and does not support arbitrary folding or expansion of contexts at any position. Interestingly, Kuratov et al. (2025) investigated the capacity of LLMs to encode long contexts into a single embedding, demonstrating minimal information loss for sequences up to 1500 tokens. Their work examines the extent to which information can be compressed into a single embedding, offering a complementary perspective to REFRAG, which is designed for decoding from multiple compact embeddings within the standard decoder architecture.

**Compressive transformer.**    Rae et al. (2020) first introduced the compressive transformer, which compresses the KV cache to reduce memory requirements for long-context applications. However, this approach only decreases KV cache memory usage, does not improve time-to-first-token latency, and requires training the model from scratch. Yoshida et al. (2021) extended this idea by employing recursive context compression, generating a summary hidden state for each chunk to inform the next chunk's computation. The recursive nature, however, prevents pre-computation and reuse of chunk embeddings, and does not reduce decoding latency. Chevalier et al. (2023) proposed recursive compression for documents, using compressed embeddings for prediction, similar to our method. However, their sequential compression process results in high latency when the summary vector is not cached, and their approach only supports applications where the summary token is restricted to the prefix of the language model (e.g., RAG), limiting applicability. In contrast, our work is the first to enable pre-computation of chunk embeddings and their use at arbitrary positions within the prompt, supporting diverse applications such as RAG and multi-turn conversation. Furthermore, our method learns where to apply compression, allowing for adaptive compression rates at inference time without recomputing chunk embeddings.

**Prompt compression.**    Prompt compression seeks to reduce input token length to lower latency and cost while maintaining task performance. A prominent approach is *LLMLingua*(Jiang et al., 2023),which employs coarse-to-fine, budget-controlled compression with token-level iterative refinement, achieving high compression ratios with minimal performance loss. *LongLLMLingua* (Jiang et al., 2024) extends this method to long-context scenarios, demonstrating significant cost and end-to-end speed improvements. Complementary approaches rank or prune context by estimated informativeness, e.g., *Selective Context* uses self-information to drop low-value tokens, and

sentence-level methods learn context-aware encoders for question-specific compression and faster inference Li et al. (2023); Liskavets et al. (2024). These approaches are complementary to our work and can be integrated to further reduce the latency of REFRAG.

# B ADDITIONAL DISCUSSION

**Analysis on latency and throughput improvement.** We denote the following parameters: $s$ as the context length, $o$ as the output length, $b$ as the batch size, $d$ as the dimensionality of the hidden states, $l$ as the number of layers in the decoder, and $n$ as the number of model parameters. The flop rate of the GPU is $f$, and the high bandwidth memory of the GPU is $m$ and we use the compression rate of $k$ in our encoder. We assume that all our chunk embeddings are precomputed and cached. The model is loaded with bfloat16 precision. We focus our analysis on LLaMA-2-7B model. The results should be generalizable to other models. We use the following metrics: TTFT which is the latency for the system to generate the first token; TTIT which is the time that it takes to generate iterative token after the first token; Throughput which is the number of tokens that are generated from the system in a unit time. Table 6 shows that with short context length $s$ we are able to achieve $k\times$ acceleration in TTFT and up to $k\times$ acceleration in throughput. With longer context length $s$, we are able to achieve up to $k^2\times$ acceleration in both TTFT and throughput. The details on the latency and throughput calculation are in Section C.4.

**Empirical verification of latency/throughput improvement.** Figure 2 shows the empirical measurement of the acceleration of REFRAG compared with CEPE, a previous work that achieves significant acceleration in inference (Yen et al., 2024). Under the context length of 16384 (i.e., mid-to-long context), REFRAG achieves $16.53\times$ acceleration in TTFT with cache and $8.59\times$ without cache. Both higher than CEPE (i.e., $2.01\times$ and $1.04\times$ acceleration respectively) while having better model performance (see Table 1). With longer context, we are able to achieve up to $32.99\times$ acceleration in TTFT. The reason why we get such acceleration even without cache is that the encoder is light-weight (e.g., Roberta-large is 355M-sized) and the chunks are processed parallel without attending to each other. In terms of TTIT, we achieve $3\times$ acceleration in long context scenario in both cached and not cached scenarios. This is expected since they have the same number of KV caches to attend to. However, CEPE is worse than original LLaMA in TTIT since it require the additional computation of KV cache projection in the inference time. Overall we achieve upto $6.78\times$ and $6.06\times$ acceleration in throughput much higher than CEPE in the long context scenario.

| | Acceleration/Save | Short $s$ | Long $s$ |
|---|---|---|---|
| **KV cache memory** | $\frac{ks+ko}{s+ko}$ | $1 \sim k\times$ | $k\times$ |
| **TTFT** | $\frac{k^2(6ds+s^2)}{6dsk+s^2}$ | $k\times$ | $k^2\times$ |
| **TTIT** | $\frac{2dlbsk+nk+2dlbok}{2dlbs+nk+2dlbok}$ | $1\times$ | $k\times$ |
| **Throughput** | $\frac{k*\text{TTFT}+k*\text{TTIT}}{\text{TTFT}+k\text{TTIT}} \sim \frac{k^2*\text{TTFT}+k^2*\text{TTIT}}{\text{TTFT}+k*\text{TTIT}}$ | $1 \sim k\times$ | $k \sim k^2\times$ |

Table 6: The acceleration in latency/save in memory of REFRAG compared to the original LLaMA model.

## B.1 MODELING REFRAG SELECTIVE COMPRESSION

In this section, we introduce selective token compression, based on the hypothesis that different context segments contribute unequally to answer prediction. Less critical segments are compressed, while essential ones remain intact, as illustrated in Fig. 5. We employ RL to train a policy that optimally determines which segments to compress.

To enable selective compression, we continue pretraining the encoder and decoder to process a combination of token and chunk embeddings. Given a context of $s$ tokens $x_1, \ldots, x_s$, chunked into $L$ fixed-length chunks $C_1, \ldots, C_L$, we achieve a compression fraction of $1 - p$ by randomly selecting $T' := pL$ chunks to remain uncompressed for the decoder. This pretraining allows the model to effectively handle mixed inputs at arbitrary positions, which is essential for the subsequent RL policy learning.

We sequentially pick $T'$ chunk indices $l = \{l_j\}_{j=1}^{T'}$, where $l_t \in [L]$. The input arrangement is $E(x, \{l_j\}_{j=1}^{T'}) = \{E_1, \ldots, E_L\}$, with $E_i = \mathbf{e}_i^{\text{cnk}}$ if $i \notin \{l_j\}_{j=1}^{T'}$ (compressed), and $E_i = \{\mathbf{e}_{k*i}, \ldots, \mathbf{e}_{k*i+k-1}\}$ if $i \in \{l_j\}_{j=1}^{T'}$ (uncompressed). This arrangement is input to the decoder $\mathcal{M}_{\text{dec}}$ to predict $x_{s+1:s+o}$. The decoder's auto-regressive property is maintained, and compression can be applied at any position within the input, not just at the beginning. Within our selective compression framework, the objective is to choose $T'$ chunks from $L$ total chunks to maximize a specified reward. Formally, this can be expressed as the following combinatorial optimization problem:

$$\text{Given} \quad [L] := \{1, 2, \ldots, L\},$$
$$\max_{l \subseteq [L]} \quad r(x, l)$$
$$\text{s.t.} \quad |l| = T'$$

This problem is non-differentiable due to its discrete nature, and exact solutions are NP-hard. Consequently, prior work has proposed greedy approaches that incrementally construct solutions by modeling the task as a sequential decision-making problem (Dai et al., 2017; Bello et al., 2017). These studies show that such greedy formulations enable the use of RL to achieve near-optimal solutions and generalize well across diverse settings. Motivated by these findings, we adopt a sequential formulation for selective compression and employ RL to train an effective policy (see Section 2).

We learn a policy network $\pi_\theta$ that takes chunk embeddings $\{\mathbf{c}_i\}_{i=1}^{L}$ and sequentially selects $T'$ chunk indices $l_1, \ldots, l_{T'}$, where $l_t \in [L]$. At stage $t$, the policy samples from:

$$\pi_\theta(l_t = i | x, \{l_j\}_{j=1}^{t-1}) := \pi_\theta(l_t = i | \{\mathbf{c}_j\}_{j=1}^{L}, \{l_j\}_{j=1}^{t-1}) = \frac{\exp(\mathbf{s}_i - \mathbf{n}_i)}{\sum_{j=1}^{L} \exp(\mathbf{s}_j - \mathbf{n}_j)} .$$

where $\mathbf{n}_j = \infty$ iff $j \in \{l_i\}_{i=1}^{t-1}$ and 0 otherwise[4]; $\mathbf{s} = g_\theta(\{\mathbf{c}_i\}_{i \in [L], i \notin \{l_j\}_{j=1}^{t-1}})$ is the output of a two-layer transformer network over chunk embeddings, producing logit $\mathbf{s}_i$ for each chunk. In practice, we reuse chunk embeddings $\{\mathbf{c}_i\}_{i=1}^{L}$ as transformer input and do not recompute logits $\mathbf{s}_i$ after each selection, as state changes have minimal impact and this improves training speed.

We use GRPO (Shao et al., 2024) style baseline to use grouped reward as baseline to reduce variance and to minimize contamination across different segment prediction task. Specifically, for each $x$ we randomly select $G$ number of length $T'$ action sequences $\{l^{(i)}\}_{i=1}^{G}$ . We have the following objective:

$$\mathcal{J}_\theta = \frac{1}{G} \sum_{i=1}^{G} \mathbb{E}_{\substack{x \sim P(\mathcal{X}), \\ \{l^{(i)}\}_{i=1}^{G} \sim \pi_\theta([L]|x)}} \frac{1}{T'} \sum_{t=1}^{T'} \min \left[ \frac{\pi_\theta(l_t^{(i)}|x, \{l_j^{(i)}\}_{j=1}^{t-1})}{\pi_{\theta_{\text{old}}}(l_t^{(i)}|x, \{l_j^{(i)}\}_{j=1}^{t-1})} A_t^{(i)}, \text{clip}\left( \frac{\pi_\theta(l_t^{(i)}|x, \{l_j^{(i)}\}_{j=1}^{t-1})}{\pi_{\theta_{\text{old}}}(l_t^{(i)}|x, \{l_j^{(i)}\}_{j=1}^{t-1})}, 1 - \epsilon, 1 + \epsilon \right) A_t^{(i)} \right]$$

$$(1)$$

where $\epsilon$ is the clipping hyperparameter in PPO (Schulman et al., 2017) for stable training, $\theta$ is the current policy and $\theta_{\text{old}}$ is the policy fro the previous iteration, $A_t$ is the advantage function. We define our advantage function using the negative log-perplexity on the $o$ tokens $x_{s+1:s+o}$:

$$r_i = r\left(x, \{l_j^{(i)}\}_{j=1}^{T'}\right) = -\mathcal{M}_{\text{dec}}\left(x_{s+1:s+o} | E(x, \{l_j^{(i)}\}_{j=1}^{T'})\right) .$$

We compute the advantage function following GRPO as:

$$A_t^{(i)} = \frac{r_i - \text{mean}\left(\{r_i\}_{i=1}^{G}\right)}{\text{std}\left(\{r_i\}_{i=1}^{G}\right)} .$$

---

[4]We adopt the masking mechanism from Pointer Networks (Bello et al., 2017) to constrain the action space.

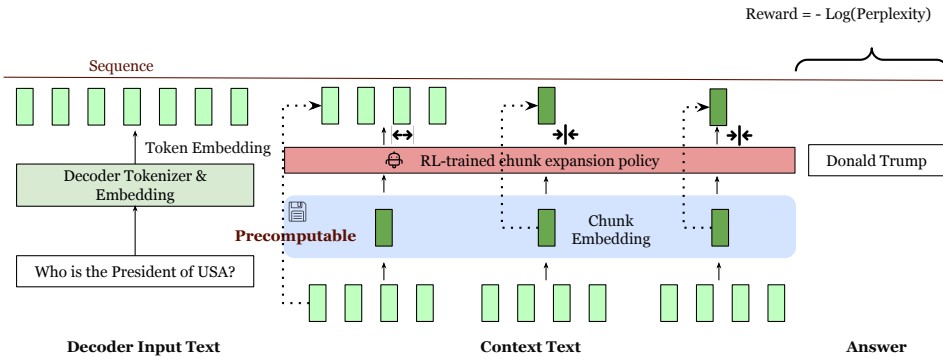

Figure 5: A demonstration of selective token compression. For all chunks, by default, we compress them to a single token, while for crucial chunks, we expand them.

## C  ADDITIONAL DETAILS ON EXPERIMENTAL SETTINGS

### C.1  ADDITIONAL DETAILS ON BASELINES

All baseline models are based on the LLaMA-2-7B model (Touvron et al., 2023), unless otherwise specified, to ensure a fair comparison since the previous methods are trained based on this model.[5] We do provide results on other encoder-decoder combinations in our ablation experiments (see Section 4.1). Each data point contains $T = 4096$ tokens, where the first $s = 2048$ tokens are referred to as the context tokens, and the remaining $o = 2048$ tokens are the output tokens, such that $s + o = T$. We evaluate the perplexity on $x_{s+1:s+o}$ in this section.

**LLAMA-NO CONTEXT:** The original pre-trained LLaMA model evaluated directly on $x_{s+1:s+o}$ with only $x_{s+1:s+o}$ as input.

**LLAMA-FULL CONTEXT:** Similar to the LLAMA-NO CONTEXT, we evaluate the perplexity on $x_{s+1:s+o}$; however, we also input the whole sequence to the model, including the context tokens, i.e., $x_{1:T}$. Therefore, the perplexity of this model is expected to be lower than LLAMA-NO CONTEXT. The perplexity of this model serves as a reference, showing the upper bound of the performance of our model.

**LLAMA$_K$:** Similar to the LLAMA-FULL CONTEXT, we pass last $K$ tokens $x_{s_K:s}$ in addition to $x_{s+1:s+o}$ to compute perplexity in $x_{s+1:s+o}$. The performance of LLAMA$_K$ falls between LLAMA-NO CONTEXT and LLAMA-FULL CONTEXT, making it a strong baseline for comparison with REFRAG when the number of context tokens is matched.

**CEPE:** A memory-efficient long-context model modified from the LLaMA model (Yen et al., 2024). The model architecture is similar to T5. We feed $x_{1:s}$ into their encoder model and evaluate the perplexity on the output tokens $x_{s+1:s+o}$. CEPED refers to its instruction fine-tuned variant.

**LLAMA-32K:** A fine-tuned version of the original LLaMA-2 7B model that extends the context length from the original 4K to 32K.

**REPLUG:** A retrieval-augmented language modeling framework that uses different retrieved contexts to perform ensemble generation. We use REPLUG to refer to applying this framework on the LLaMA pre-trained model, REPLUG$_{Chat}$ to refer to applying this framework on the LLaMA chat model (i.e., instruction fine-tuned), and REPLUG$_{FT}$ to refer to applying it on the LLaMA model fine-tuned on the downstream tasks (see Section 5).

**REFRAG:** Our approach is illustrated in Fig. 1. We use RoBERTa-large (Liu et al., 2019) as the encoder, feeding $x_{1:s}$ tokens and evaluating the perplexity on the output tokens $x_{s+1:s+o}$. We use

---

[5]Unless specified, we use the pre-trained checkpoint. The reason of choosing this model is that existing baselines (Yen et al., 2024; Shi et al., 2024) adapts LLaMA-2-7B. If we use other base model, we will have to retrain their model for fair comparison. We show the effectiveness of our training recipe in Table 3.

REFRAG$_k$ to denote our model with compression rate of $k$. We use REFRAG$_{RL}$ to refer to the model with selective compression using our RL policy.

## C.2 ADDITIONAL DETAILS ON HYPERPARAMETERS AND EXPERIMENTAL SETTINGS FOR CPT

**Hyperparameters.** For reconstruction stage, we use a peak learning rate of $2e-4$ since we only train the encoder model. For the next paragraph prediction we use a peak learning rate of $5e-5$ since we train all the parameters in the model, including the decoder parameters. For all the instruction-tuning tasks, we use the peak learning rate of $2e-5$. We use a $4\%$ linear warm-up stage for learning rate, AdamW optimizer (Loshchilov & Hutter, 2019), cosine learning rate scheduler and a batch size of 256 for all the experiments. For the projection layer, we use a 2-layer multi-layer perception (MLP) with an hidden size that is equivalent to the output size (i.e., 4096 for LLaMA-2-7B). For both tasks we train our model for 4 epochs on the dataset using the curriculum learning schedule (see Fig. 6).

**Computational Resources.** We train all our models in Bfloat16 precision. We adopt Fully Sharded Data Parallel (FSDP) for all the experiments and train our model on 8 nodes with 8 H100 cards on each node.

**Evaluation metrics in RAG.** Table 7 provides a summarization of the evaluation metrics we use for each dataset in RAG experiments.

**Experimental setting for fine-tuning model to take a combination of token and chunk embedding as input.** We continue the model training from the continual pre-training checkpoint. To fine-tune the model, we set $p = 0.1$ (i.e., compression $90\%$ of the chunks) and randomly select $pL$ chunks to keep their original token in the decoder. The input arrangement is the same as what we describe in Section 2.

| Dataset | Metric |
|---|---|
| OpenAssistant Conversations | F1 |
| CommonsenseQA | Accuracy |
| MathQA | Accuracy |
| Web Questions | Exact Match |
| WikiQA | F1 |
| Yahoo! Answers QA | F1 |
| FreebaseQA | Exact Match |
| MS MARCO | F1 |
| PubMedQA | Exact Match |
| QuaRel | Accuracy |
| GSM8K | Exact Match |
| StrategyQA | Exact Match |
| MMLU | Accuracy |
| BoolQ | Exact Match |
| SIQA | Accuracy |
| PIQA | Accuracy |
| HellaSwag | Accuracy |
| Winogrande | Accuracy |
| TriviaQA | Exact Match |
| FEVER | Exact Match |
| NQ | Exact Match |

Table 7: Metrics used for each dataset in RAG experiments in Table 4

## C.3 CURRICULUM LEARNING DATA MIXTURE

Table 8 presents the number of data points used at each training stage of our model. We employ a geometric sequence for each type of data point, based on the intuition that training should begin with

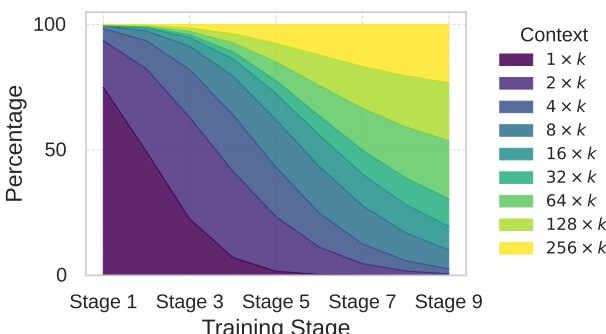

Figure 6: The data mixture in curriculum learning during the training.

| Factor | Stage 1 | Stage 2 | Stage 3 | Stage 4 | Stage 5 | Stage 6 | Stage 7 | Stage 8 | Stage 9 | Summation |
|---|---|---|---|---|---|---|---|---|---|---|
| $1 \times 8$ | 1333 | 445 | 148 | 49 | 16 | 6 | 2 | 1 | 0 | 2000 |
| $2 \times 8$ | 333 | 298 | 267 | 238 | 213 | 191 | 171 | 153 | 137 | 2000 |
| $4 \times 8$ | 83 | 102 | 126 | 156 | 193 | 238 | 293 | 362 | 447 | 2000 |
| $8 \times 8$ | 20 | 35 | 61 | 106 | 185 | 324 | 565 | 985 | 1719 | 4000 |
| $16 \times 8$ | 5 | 11 | 23 | 48 | 103 | 220 | 468 | 997 | 2125 | 4000 |
| $32 \times 8$ | 1 | 3 | 7 | 19 | 50 | 133 | 353 | 939 | 2496 | 4000 |
| $64 \times 8$ | 1 | 3 | 9 | 25 | 73 | 212 | 618 | 1802 | 5259 | 8000 |
| $128 \times 8$ | 1 | 3 | 9 | 25 | 73 | 212 | 618 | 1802 | 5259 | 8000 |
| $256 \times 8$ | 1 | 3 | 9 | 25 | 73 | 212 | 618 | 1802 | 5259 | 8000 |

Table 8: The geometry curriculum learning scheduling. The whole training is split into 9 stages. In each stage, we have a combination of different data (e.g., 1X8 means reconstructing 8 tokens, 2X8 means reconstructing 16 tokens). For each type of data, the number of samples in each stage is determined by a geometric sequence which sums up to the total number of samples in the last column. As training proceeds, the data mixture has more and more longer sequences.

a greater proportion of easier examples and gradually introduce more challenging ones as training progresses. The right-most column indicates the total number of data points for each type. We allocate more data points to longer-context examples to encourage the model to focus on learning more difficult tasks.

## C.4 DETAILED CALCULATION OF ACCELERATION IN LATENCY AND THROUGHPUT OF OUR MODEL

In this section, we provide a detailed analysis of the TTFT and generation latency for the LLaMA-2 model. We denote the following parameters: $s$ as the context length, $o$ as the output length, $b$ as the batch size, $d$ as the dimensionality of the hidden states, $l$ as the number of layers in the decoder, and $n$ as the number of model parameters. The flop rate of the GPU is $f$, and the high bandwidth memory of the GPU is $m$. The model is loaded with bfloat16 precision. Note that the inference/training is FlashAttention-compatible and our implementation already uses FlashAttention. We focus our analysis on LLaMA-2-7B model. The results should be generalizable to other models.

**TTFT: Computationally Bounded Analysis**  Existing work (Liu et al., 2025) has shown that the TTFT latency is primarily limited by computation. The primary computations in each layer of LLaMA-2 involve attention calculations and feedforward layers. We follow the analysis in (Liu et al., 2025) to calculate the TTFT. Note that each operation involves both a multiplication and an addition, hence we multiply the flop count by 2.

- **Attention Calculation:**
    - *QKV Projection:* Transforms input from $[b, s, d]$ to $[d, 3d]$, requiring $6bsd^2$ flops.
    - *Attention Score Calculation:* $QK^T$ operation from $[b, h, s, d/h] \times [b, h, d/h, s]$, requiring $2bds^2$ flops.

- *Attention Output Calculation:* Weighted average of the value hidden state, $[b, h, s, s] \times [b, h, s, d/h]$, requiring $2bds^2$ flops.
    - *Output Projection:* $[b, s, d] \times [d, d]$, requiring $2bsd^2$ flops.

    The total flops for attention is $8bsd^2 + 4bds^2$.

- **Feedforward Layer:** In LLaMA-2-7B, the MLP layer first projects to $2.6875d$ with a gated function and then back to $d$. Each projection requires $5.375bsd^2$ flops. With three such operations, the total is $16.125bsd^2$.

- **Total Computation per Layer:** Summing the above, each layer requires approximately $24bsd^2 + 4bds^2$ flops.

For a sequence length $s$, number of layers $l$, and batch size $b$, the total computation for pre-fill is $(24d^2 + 4ds)lbs$. Given the flop rate $f$, the latency for pre-fill is dominated by computation, yielding a final latency of $\frac{(24d^2 + 4ds)lbs}{f}$.

**Generation analysis: Memory bounded Analysis** For generation latency, existing work have shown that the generation process is memory bounded (Shi et al., 2025) which requires transferring KV cache and model parameter to high-bandwidth memory, we analyse the data transfer latency as follows:

- **Memory Latency:**
    - *KV Cache Data:* Requires $4dlb(s + o)$ bytes (bfloat16 uses 2 bytes per number, and there are separate key/value copies).
    - *Model Parameters:* Require $2n$ bytes.

    The data transfer latency to high-bandwidth memory is $\frac{2n + 4dlb(s+o)}{m}$.

**Throughput Calculation** The throughput, defined as the number of tokens generated per unit time, is given by:

$$\text{Throughput} = \frac{bo}{\text{TTFT} + \text{DL}}$$

where DL is the data latency.

| | Before | After |
|---|---|---|
| **KV cache memory** | $4dlb(s + o)$ | $4dlb\left(\frac{s}{k} + o\right)$ |
| **TTFT** | $\frac{(24d^2 + 4ds)lbs}{f}$ | $\frac{(24d^2 + 4d\frac{s}{k})lb\frac{s}{k}}{f}$ |
| **TTIT** | $\frac{2n + 4dlb(s+o)}{m}$ | $\frac{2n + 4dlb\left(\frac{s}{k} + o\right)}{m}$ |
| **Throughput** | $\frac{bo}{\text{TTFT}_{\text{before}} + \text{TTIT}_{\text{before}}}$ | $\frac{bo}{\text{TTFT}_{\text{after}} + \text{TTIT}_{\text{after}}}$ |

Table 9: Comparison of KV cache memory usage, TTFT, generation latency and throughput between the original LLaMA model and our model.

## C.5 Additional details on empirical measurement of latency and memory improvement in Fig. 2, Fig. 9 and Fig. 8

We measure the latency and memory usage in a controlled environment which aims to reduce other environmental factors that could make certain method advantageous.

To this end, our implementation uses the same modelling file which means different baselines share the same hyper-parameter and acceleration (e.g., flash-attention). Therefore, we restrict the factors that affect the resource usage only among the model designs. We use the batch size of 1 and use a single A100 card to measure the system performance.

In a typical AI web search engine, the retrieval component can take 50-300 ms to get multiple documents/chunks from a index containing billions of webpages. For a context length of $\sim 32k$, the decoder prefill takes $1.5 - 2s$. REFRAG reduce the decoder prefill stage by $16x$ for $k = 16$. The

(optional, dynamic) RL selective-expansion module contributes $\leq 2$ ms per request for deciding which chunks to expand; its overhead is negligible. The main effect of expansion is that a small fraction of chunks are restored to tokens, increasing prefill length. The effective TTFT speedup is therefore scaled by $\frac{k}{1-p+kp}$ where we expand a small fraction $p$ of chunks.

## C.6 TRAINING COST

REFRAG requires three stages of training: (i) Reconstruction pretraining (20B tokens) takes $\sim 2000$ H100 GPU-hours, (ii) CPT pretraining (20B tokens): $\sim 2000$ H100 GPU-hours, and (iii) RL selective expansion takes $\sim 300$ H100 GPU-hours. The RL policy is intentionally lightweight: a 2-layer MLP over chunk embeddings with a small discrete action space.

## D ADDITIONAL EXPERIMENTAL RESULTS

**Sparse attention across different retrieved passages.**    We retrieve 200 passages using the query "how bruce lee died" from our retrieval corpus. We choose 5 passages that are different from each other (Table 10) to simulate the de-duplication process in real RAG applications. We concatenate these 5 passages and feed it to LLaMA-2-7B-Chat model to see the attention values between different tokens. Figure 7 shows that the attention values for tokens within each passages are significantly larger than attention values for tokens in different passages which suggests redundancy in the current attention computation for RAG applications.

**Additional results in latency measurement.**    Figure 9 and Fig. 8 shows the latency comparison of different models when using $k = 8$ and $k = 32$ compression rate for REFRAGrespectively.

**Ablation study result for curriculum learning.**    Table 11 shows the necessity of curriculum learning to the success of reconstruction task.

**Ablation study result for reconstruction task.**    Table 12 shows the performance comparison in CPT with and without continuing from reconstruction task.

**Ablation study result for the advantage of RL.**    Table 13 shows the advantage of using our selective compression policy via RL compared to using a lower compression rate. The negative next-paragraph log-perplexity used in REFRAG produces the lowest variance across rollouts and converges stably in all runs. As shown in Fig. 3, the learned policy consistently outperforms heuristic baselines across compression rates, and we observe no hypersensitivity to reward scaling.

**Ablation study result of different compression rates.**    Figure 10 shows the loss trajectory for different compression rate of REFRAG.

**Ablation study result of different combination of encoder and decoder models.**    Figure 11 shows the performance of CPT with different combination of encoder and decoder models. Table 3 shows the performance on LLaMA-3.1-8B and LLaMA-3.2-3B model. The perplexity of REFRAG remains close to the full context perplexity under more modern architecture

**Additional results in RAG.**    Table 16 shows the performance of different baselines under the same number of context. The performance of our model is similar to other methods, in other words no model significantly outperforms others. Table 15 shows the performance of REFRAG under different number of context for strong retriever setting.

**Demonstration of generated summary for Arxiv and Pubmed articles.**    Table 20 and Table 19 shows the ground true abstract for different articles and the generated summary from REFRAG. These results complement the perplexity results we have shown in CPT and accuracy/F1 performance we have shown in RAG and other applications.

Table 10: The 5 retrieved passages for the query "how bruce lee died".

|  | Content |
|---|---|
| P0 | "Water is necessary to survive, but as we all know, sometimes too much of a good thing (even water) can be harmful. In 2022, a group of kidney specialists from Madrid, Spain, revisited the death of Kung Fu legend Bruce Lee and concluded that water intoxication was the most likely cause of his untimely death. Bruce Lee, the martial arts legend and iconic figure in the history of cinema, died on July 20, 1973, at the young age of 32. The official cause of death at the time was reported as a probable drug reaction and classified as "death by misadventure." Hours before his death, Lee complained of a headache while visiting a fellow actress Betty Ting Pei at her apartment. She gave him one of her own prescription painkillers (one that contained aspirin and meprobamate), and he laid down to take a nap. He never woke up and was unable to be resuscitated even after being transferred to a Hong Kong hospital. In the years since Lee's death, many theories have been put forward as to the true cause of his passing. These theories include murder by gangsters or a jilted lover, a family curse, epilepsy, heatstroke, and possibly |
| P1 | Bruce Lee May Have Died From Drinking Too Much Water, Claims Study The 'Enter The Dragon' actor, who helped bring martial arts into popular culture, died in July 1973 at the age of 32. American martial arts legend and actor Bruce Lee might have died from drinking too much water, scientists have claimed in a new study. The 'Enter The Dragon' actor, who helped bring martial arts into popular culture, died in July 1973 at the age of 32 from cerebral oedema, a swelling of the brain. At the time, doctors believed the brain swelling was due to a painkiller. The oedema, according to a group of researchers, was brought on by hyponatraemia. In their study, which was published in the Clinical Kidney Journal, the researchers proposed that Bruce Lee died because his kidneys were unable to eliminate extra water. The findings are very different from old theories about how died, such as those regarding gangster assassination, jealous lover poisoning, curses, and heatstroke. According to scientists, the actor may have died from hyponatraemia, which develops when the body's sodium levels get diluted as a result of consuming too much water. The cells in the body, particularly those in the brain, |
| P2 | circumstances, you're bound to get some truly insane conspiracy theories, and there are plenty about Bruce Lee. The crazy Bruce Lee murder theories Producer Raymond Chow made a big mistake after Bruce Lee's death. Hoping to protect Lee's image, Chow's production company claimed the actor died at home with his wife, Linda. But once the press found out the truth, the tabloids got going. In fact, a lot of people pointed the finger at Betty Ting Pei, claiming she was responsible for Lee's death and that perhaps she'd even poisoned him. Unfortunately, that wasn't the only rumor involving murder. One of the most popular theories says other martial artists were angry at Lee for teaching their secrets to Westerners, so they decided to bump him off. Some say ninjas were responsible, and others claim Lee was killed with the "Dim Mak," a mythical martial arts move that supposedly kills a victim with one fateful blow. Others believe he was killed after refusing to pay protection money to the Triads, while others claim the Mafia did the deed because Lee wouldn't let them control his career. The more mystical conspiracy theorists even say there's a family curse that took the life |
| P3 | Bruce Lee complained of a headache, was given an Equagesic — a painkiller that contains both aspirin and the tranquilizer meprobamate — and went down for a nap. He never woke up. His death was said to be an allergic reaction to the tranquilizer resulting in a cerebral edema (he had suffered a previous edema months before), though others claim his death was due to a negative reaction to cannabis, which Lee consumed regularly to reduce stress. Because he was so young, news of his death invited wild media speculation, from murder to a family curse. 5. Brandon Lee Sadly, Bruce Lee's son Brandon also died young, at age 28, and also under strange circumstances. While filming the horror film The Crow, Lee was accidentally killed by a prop gun that, due to a malfunction in a previous scene, was accidentally loaded with a dummy bullet and a live primer. When the gun was fired, the bullet was ejected with virtually the same force as if loaded with a live round. Lee was hit in the abdomen and died in surgery later that day, on March 31, 1993. Like his father, Brandon's abrupt death fed rumors. Conspiracy theorists believe Illuminati |
| P4 | Bruce Lee moved to a house in Hong Kong's Kowloon Tong district, it was said that the building suffered from bad feng shui. According to Lee biographer Bruce Thomas, the house's two previous owners had financial issues, and the building "faced the wrong way," and had disturbed natural winds. To fix this problem, a feng shui adviser ordered a mirror to be put on the roof. This was supposed to deflect the bad energy, but the mirror was knocked off during a typhoon. Ominously, Lee died just two days after the charm was blown away. While some of Lee's neighbors apparently linked the two events at the time, the problem with this theory is that feng shui is nothing but a superstition. There's no scientific evidence for any of its tenets, including qi. At most, feng shui could be regarded as a kind of art. Lee's death after the loss of his mirror is a simple coincidence. Moreover, Lee died in Betty Ting's apartment, not in his own house. 2. Murder The abruptness of Bruce Lee's death, combined with his extraordinary fitness, made some fans wonder whether something more sinister was at work. People who believe that Lee was murdered |

Table 11: Performance comparison on reconstruction task with and w/o curriculum learning. Log-Perplexity is reported as average of Arxiv and Book domain.

|  | P16 | P32 | P128 | P2048 ↓ |
|---|---|---|---|---|
| LLaMA-Full Context | 1.397 | 0.734 | 0.203 | 0.021 |
| LLaMA-No Context | 3.483 | 2.981 | 2.249 | 1.590 |
| REFRAG w/o curriculum | 3.719 | 3.098 | 2.272 | 1.599 |
| REFRAG with curriculum | 0.669 | 0.451 | 0.230 | 0.135 |

Table 12: Performance comparison on continual pre-training task with and w/o continued from reconstruction task. Log-Perplexity is reported as average of Arxiv and Book domain.

|  | P16 | P32 | P128 | P2048 ↓ |
|---|---|---|---|---|
| LLaMA-Full Context | 1.448 | 1.458 | 1.464 | 1.449 |
| LLaMA-No Context | 3.483 | 2.981 | 2.249 | 1.590 |
| REFRAG w/o reconstruction | 3.272 | 2.789 | 2.119 | 1.544 |
| REFRAG with reconstruction | 2.017 | 1.837 | 1.632 | 1.453 |

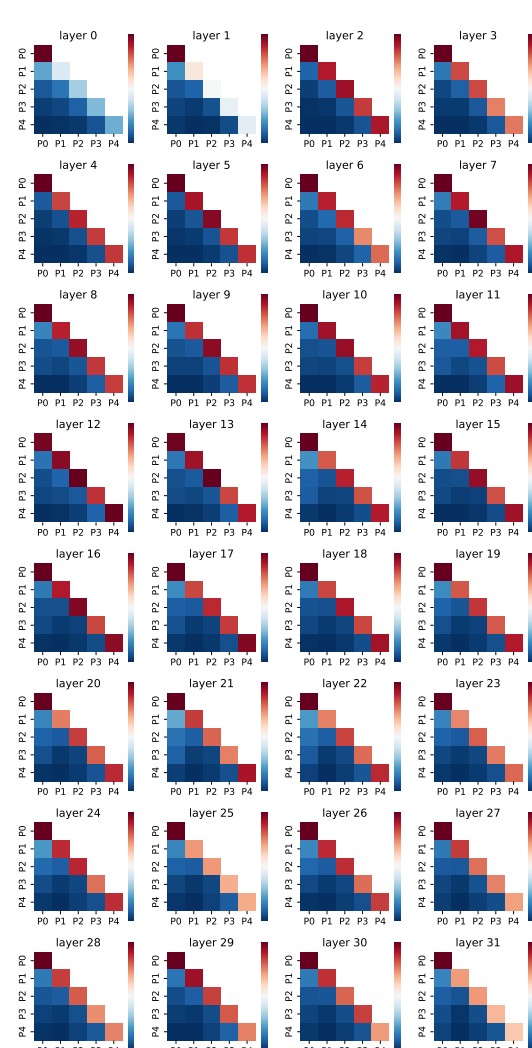

Figure 7: Attention value visualization for different retrieved passages for different layers for LLaMA-2-7B-Chat model. The diagonal values are the averaged attention value for tokens within each passage while the off-diagonal values are the averaged attention value between tokens from different passages. The detail of retrieved passages is in Table 10.

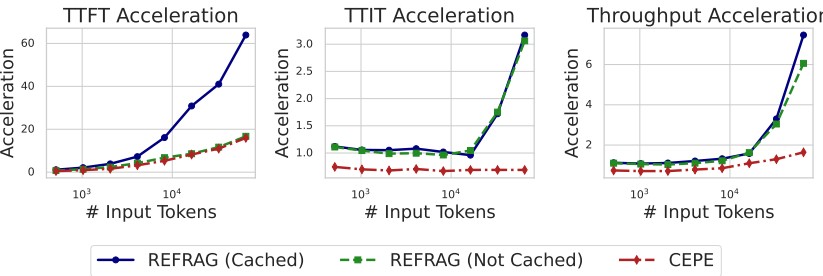

Figure 8: Empirical verification of inference acceleration of REFRAG with $k = 32$.

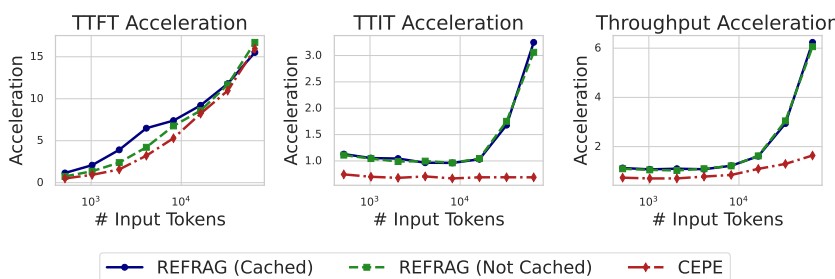

Figure 9: Empirical verification of inference acceleration of REFRAG with $k = 8$.

Table 13: The performance of REFRAG under the same compression rate with full compression (i.e., REFRAG$_8$) and selective compression (i.e., REFRAG$_{16+RL}$).

| | Compression Rate | Arxiv | | | Book | | | PG19 | | | ProofPile | | |
|---|---|---|---|---|---|---|---|---|---|---|---|---|---|
| | | P512 | P1024 | P2048 | P512 | P1024 | P2048 | P512 | P1024 | P2048 | P512 | P1024 | P2048 ↓ |
| Context Length=2048 | | | | | | | | | | | | | |
| REFRAG$_8$ | 8 | 1.124 | 1.091 | 1.062 | 1.905 | 1.868 | 1.844 | 1.996 | 1.956 | **1.927** | 0.997 | 0.952 | 0.916 |
| REFRAG$_{16+RL}$ | 8.258 | **1.118** | **1.090** | 1.062 | **1.878** | **1.856** | **1.840** | **1.978** | **1.952** | 1.930 | **0.992** | **0.951** | 0.916 |
| Context Length=4096 | | | | | | | | | | | | | |
| REFRAG$_8$ | 8 | 1.098 | 1.065 | 1.042 | 1.895 | 1.860 | 1.837 | 1.989 | 1.950 | 1.922 | 0.965 | 0.923 | 0.894 |
| REFRAG$_{16+RL}$ | 8.0157 | **1.065** | **1.048** | **1.033** | **1.851** | **1.837** | **1.828** | **1.952** | **1.934** | **1.918** | **0.932** | **0.905** | **0.883** |

Table 14: Performance on multi-turn RAG tasks with different number of passages.

| | REFRAG | | | LLAMA$_{FT}$ | | |
|---|---|---|---|---|---|---|
| # Passages | ORConvQA | QReCC | TopiOCQA ↑ | ORConvQA | QReCC | TopiOCQA ↑ |
| 0 | 19.27 | 15.32 | 28.19 | 19.16 | 15.49 | 28.22 |
| 5 | 20.18 | 17.37 | **28.24** | 19.65 | **18.71** | 27.08 |
| 8 | **20.52** | 17.60 | 28.17 | 16.87 | 18.05 | 25.36 |
| 10 | 19.67 | 17.41 | 27.62 | 15.72 | 17.42 | 23.60 |

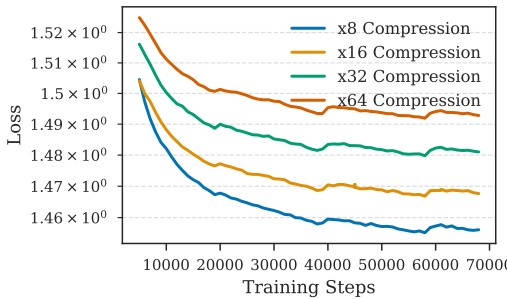

Figure 10: Training trajectory for our model with different compression rate.

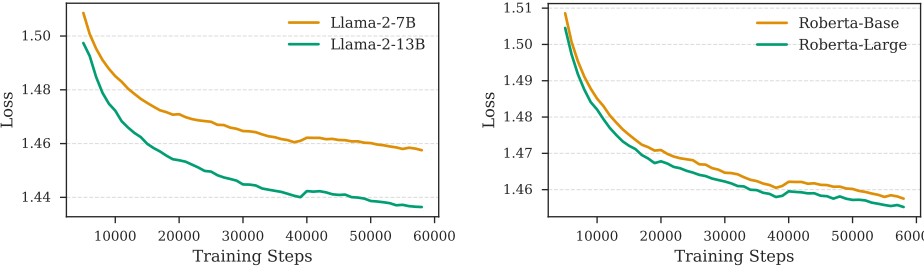

Figure 11: Training trajectory for different encoder and decoder combinations. On the left, we have two different decoder the Roberta-Base encoder. On the right we have two different encoder for LLaMA-2-7B decoder model.

Table 15: Performance of our model under compression rate of 16 with different number of retrieved passages in RAG under the strong retriever scenario.

| # Passages | MMLU | NQ | FEVER | WebQA | FreebaseQA | CommonsenseQA | ECQA | StrategyQA | HellaSwag | SIQA | PIQA ↑ |
|---|---|---|---|---|---|---|---|---|---|---|---|
| 0 | 48.07 | 18.73 | 65.80 | 34.67 | 60.20 | 89.18 | 87.42 | 68.89 | 43.72 | 67.25 | 70.18 |
| 1 | 50.49 | 21.39 | 69.46 | 37.33 | 68.06 | 86.60 | 89.40 | 80.00 | 43.26 | 68.17 | 70.08 |
| 3 | 50.49 | 22.01 | 66.02 | 38.67 | 71.01 | 89.18 | 95.36 | 71.11 | 45.50 | 68.73 | 71.44 |
| 5 | 50.62 | 23.00 | 66.07 | 41.33 | 72.48 | 91.75 | 96.03 | 75.56 | 45.48 | 68.17 | 71.38 |
| 8 | 50.29 | 22.96 | 66.59 | 38.67 | 73.46 | 92.27 | 94.70 | 75.56 | 45.23 | 68.94 | 71.38 |
| 20 | 51.01 | 24.30 | 67.77 | 40.00 | 75.18 | 91.75 | 98.01 | 75.56 | 45.09 | 68.53 | 71.00 |
| 50 | 51.08 | 24.76 | 69.39 | 40.00 | 75.92 | 91.75 | 97.35 | 75.56 | 44.78 | 67.81 | 69.97 |
| 80 | 50.42 | 24.15 | 68.83 | 37.33 | 74.20 | 92.27 | 97.35 | 71.11 | 44.61 | 68.22 | 69.37 |
| 100 | 50.23 | 23.99 | 69.80 | 36.00 | 74.45 | 92.27 | 97.35 | 71.11 | 44.57 | 68.07 | 69.75 |

Table 16: Comparison of model performance of different models with different number of retrieved chunks for RAG. The number of contexts in all the evaluation here is 5.

| Generation | NQ | FEVER | TQA | WebQA | FreebaseQA | GSM8K | StrategyQA | BoolQ ↑ |
|---|---|---|---|---|---|---|---|---|
| LLaMA$_{FT}$ | **21.88** | 61.85 | 7.96 | 34.67 | **72.97** | 8.72 | 71.11 | **29.54** |
| CEPE | 0.05 | 60.68 | 0.01 | 0.00 | 0.25 | 0.00 | 0.00 | 56.70 |
| REPLUG | 14.96 | **71.56** | 11.01 | 25.33 | 53.32 | 4.70 | 66.67 | 3.15 |
| LLaMA-32K | 2.26 | 0.23 | 2.17 | 14.67 | 9.83 | 0.67 | 4.44 | 0.06 |
| REFRAG$_8$ | 20.86 | 63.44 | **12.37** | 38.67 | 65.60 | 11.41 | **73.33** | 3.06 |
| REFRAG$_{16}$ | 20.60 | 60.45 | 11.86 | **40.00** | 66.09 | 11.41 | 73.33 | 5.57 |
| REFRAG$_{32}$ | 21.39 | 61.97 | 12.03 | 40.00 | 67.32 | **12.75** | 68.89 | 1.80 |

| Multi-Choice | MMLU | CommonsenseQA | MathQA | ECQA | HellaSwag | SIQA | PIQA | Winogrande ↑ |
|---|---|---|---|---|---|---|---|---|
| LLaMA$_{FT}$ | **49.97** | 84.02 | 97.48 | 86.09 | 42.78 | 67.09 | 68.39 | 54.78 |
| CEPE | 26.06 | 20.62 | 24.16 | 19.87 | 24.99 | 33.57 | 49.13 | 46.96 |
| REPLUG | 47.35 | 77.84 | 99.50 | 79.47 | **49.26** | 64.99 | **71.98** | 56.04 |
| LLaMA-32K | 24.17 | 18.04 | 22.32 | 15.89 | 24.09 | 16.84 | 28.02 | 48.78 |
| REFRAG$_8$ | 49.90 | **91.24** | **99.66** | **97.35** | 45.03 | 68.27 | 70.95 | **57.22** |
| REFRAG$_{16}$ | 49.84 | 90.21 | 99.66 | 96.69 | 39.52 | **68.63** | 70.95 | 56.35 |
| REFRAG$_{32}$ | 49.84 | 91.24 | 99.50 | 97.35 | 42.71 | 68.32 | 68.72 | 56.12 |

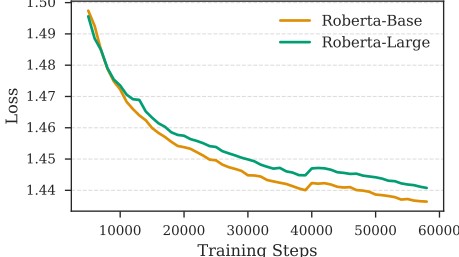

Figure 12: Training trajectory for different encoder paired with LLaMA-2-13B decoder.

Table 17: Comparison of model performance of different models with different number of retrieved passages for RAG under the weak retriever scenario.

| Generation | NQ | FEVER | TQA | WebQA | FreebaseQA | GSM8K | StrategyQA | BoolQ ↑ | (1/ # tokens) |
|---|---|---|---|---|---|---|---|---|---|
| Short context with the same latency | | | | | | | | | |
| LLaMA$_{FT}$ + 1 passage | 20.20 | 57.70 | 8.32 | 32.00 | 67.08 | 6.71 | 62.22 | **31.25** | 1× |
| REFRAG$_8$+ 8 passages | **21.22** | **63.21** | 11.77 | 42.67 | **67.57** | 8.72 | **68.89** | 3.24 | 1× |
| REFRAG$_{16}$+ 8 passages | 20.73 | 60.86 | 11.60 | 40.00 | 66.83 | 11.41 | 77.78 | 6.36 | 2× |
| REFRAG$_{32}$+ 8 passages | **21.08** | 62.65 | 11.69 | 42.67 | 66.58 | 11.41 | 68.89 | 2.35 | 4× |
| Long context | | | | | | | | | |
| LLaMA$_{FT}$ + 10 passages | 22.27 | 60.40 | 8.32 | 38.67 | 71.50 | 9.40 | 71.11 | 29.94 | 1× |
| LLaMA-32K +80 passages | 1.03 | 0.12 | 0.37 | 5.33 | 9.34 | 0.00 | 0.00 | 0.03 | |
| REFRAG$_8$ +80 passages | **22.92** | **67.87** | **12.22** | 46.67 | **71.99** | 10.07 | 68.89 | 7.19 | 1× |
| REFRAG$_{16}$ +80 passages | **22.63** | 65.07 | **12.12** | 38.67 | **71.74** | 8.72 | 68.89 | 12.05 | 2× |
| REFRAG$_{32}$ +80 passages | 21.86 | 67.24 | 11.54 | 41.33 | 70.76 | 8.72 | 66.67 | 6.30 | 4× |
| CEPED +80 passages | 0.02 | 65.18 | 0.02 | 0.00 | 0.00 | 0.00 | 0.00 | **59.33** | |
| REPLUG +80 passages | - | - | - | - | - | - | 64.44 | - | |

| Multi-Choice | MMLU | CommonsenseQA | MathQA | ECQA | HellaSwag | SIQA | PIQA | Winogrande ↑ | |
|---|---|---|---|---|---|---|---|---|---|
| Short context with the same latency | | | | | | | | | |
| LLaMA$_{FT}$ + 1 context | 48.86 | 82.99 | 99.50 | 84.77 | 42.08 | 67.91 | 67.46 | 55.49 | 1× |
| REFRAG$_8$ + 8 passages | **50.10** | **91.24** | **99.66** | 96.03 | 45.15 | 68.17 | 70.40 | **57.46** | 1× |
| REFRAG$_{16}$ + 8 passages | 49.77 | 90.21 | 99.66 | 96.69 | 39.32 | 68.73 | 70.46 | 56.43 | 2× |
| REFRAG$_{32}$ + 8 passages | **50.10** | 91.75 | 99.50 | 96.03 | 42.36 | 68.83 | 68.28 | 55.80 | 4× |
| Long context | | | | | | | | | |
| LLaMA$_{FT}$ + 10 passages | 45.20 | 83.51 | 63.42 | 85.43 | 41.43 | 67.60 | 67.36 | 54.30 | 1× |
| LLaMA-32K +80 passages | 22.01 | 18.04 | 19.97 | 16.56 | 23.69 | 23.80 | 33.19 | 48.62 | |
| REFRAG$_8$ +80 passages | 50.03 | 90.72 | **99.66** | 97.35 | 44.44 | 67.66 | 69.48 | 56.91 | 1× |
| REFRAG$_{16}$ +80 passages | 49.77 | 90.21 | 99.66 | 95.36 | 38.29 | 68.12 | 70.57 | 56.91 | 2× |
| REFRAG$_{32}$ +80 passages | 50.03 | 91.24 | 99.50 | 98.01 | 43.02 | 68.58 | 68.55 | 57.22 | 4× |
| CEPED +80 passages | 26.52 | 24.74 | 23.83 | 22.52 | 24.97 | 32.86 | 48.80 | 44.20 | |
| REPLUG +80 passages | - | - | - | 76.16 | - | 65.46 | - | 55.33 | |

- means the corresponding model has out-of-memory error.

Table 18: Performance of our model under compression rate of 16 with different number of retrieved passages in RAG under the weak retriever scenario.

| # Passages | MMLU | NQ | FEVER | WebQA | FreebaseQA | CommonsenseQA | ECQA | StrategyQA | HellaSwag | SIQA | PIQA↑ |
|---|---|---|---|---|---|---|---|---|---|---|---|
| 0 | 48.14 | 19.09 | 61.40 | 30.67 | 59.71 | 85.05 | 86.75 | 55.56 | 36.57 | 64.59 | 68.82 |
| 1 | 49.97 | 20.08 | 64.15 | 38.67 | 64.62 | 87.63 | 92.72 | 71.11 | 39.08 | 68.58 | 70.57 |
| 3 | 49.64 | 20.63 | 60.80 | 40.00 | 68.55 | 89.69 | 95.36 | 75.56 | 39.41 | 69.40 | 71.11 |
| 5 | 49.84 | 20.60 | 60.45 | 40.00 | 66.09 | 90.21 | 96.69 | 73.33 | 39.52 | 68.63 | 70.95 |
| 8 | 49.77 | 20.73 | 60.86 | 40.00 | 66.83 | 90.21 | 96.69 | 77.78 | 39.32 | 68.73 | 70.46 |
| 20 | 50.03 | 21.29 | 62.32 | 36.00 | 68.06 | 89.69 | 95.36 | 75.56 | 38.58 | 69.29 | 70.62 |
| 50 | 49.84 | 22.12 | 63.54 | 37.33 | 71.99 | 89.69 | 96.69 | 75.56 | 38.11 | 68.53 | 70.84 |
| 80 | 49.77 | 22.63 | 65.07 | 38.67 | 71.74 | 90.21 | 95.36 | 68.89 | 38.29 | 68.12 | 70.57 |
| 100 | 50.62 | 22.80 | 65.17 | 37.33 | 73.46 | 89.69 | 96.03 | 68.89 | 38.51 | 68.37 | 70.18 |

| Ground True Abstract | Generated Abstract |
|---|---|
| background : timely access to cardiovascular health services is necessary to prevent heart damages . the present study examined inequality in geographical distribution of cardiovascular health services in iran . methods : present study is a cross - sectional study conducted using demographic data from all iranian provinces ( 31 provinces ) from 2012 census by the statistics center of iran ( sci ) . the gini coefficients of ccu beds and cardiologists were used to assess equality in access to cardiovascular health services in iran . ms excel software was used to calculate gini coefficients . results : the proportions of ccu bed and cardiologist per 100,000 population were 4.88 and 1.27 , respectively ; also the gini coefficients were 0.129 and 0.045 , respectively . conclusion : descriptive statistics showed a skewness in distribution of pubic cardiovascular health services in iran , though gini coefficient revealed no significant inequality . however , equal distribution of ccu beds and cardiovascular specialists does not mean they are sufficiently available in iran . | background : this study aimed to investigate the inequality of distribution of cardiac care units ( ccu ) and cardiologists in iran . methods : this study used demographic data from national statistics collected by the central statistics of iran ( sci ) in 2012 . the number of ccu beds and cardiologists per 100,000 individuals and the number of cardiologists per 10 ccu beds were explored . the gini coefficient was applied to measure inequality . results : the mean number of ccu beds per 100,000 individuals in iran was 4.88 , which is out of a total of 31 provinces below the national mean . the mean number of cardiologists per 100,000 individuals was 1.27 and the mean number of cardiologists per 10 ccu beds was 2.6 . the gini coefficient of cardiologists in 2012 was 0.045 and the gini coefficient of ccu beds was 0.129 . conclusion : our descriptive statistics showed that there is a skewness in the distribution of pubic cardiovascular health services in iran . moreover , the equal distribution of cardiovascular health facilities such as ccu beds is not necessarily provided in iran . |
| lumbar spinal stenosis is a commonly treated with epidural injections of local anesthetics and corticosteroids , however , these therapies may relieve leg pain for weeks to months but do not influence functional status . furthermore , the majority of patients report no substantial symptom change over the repeated treatment . utilizing balloon catheters , we successfully treated with three patients who complained persistent symptoms despite repeated conventional steroid injections . our results suggest that transforaminal decompression using a balloon catheter may have potential in the nonsurgical treatment of spinal stenosis by modifying the underlying pathophysiology . | epidural injection is a common treatment for spinal stenosis . however , there is little information on the optimal management of spinal stenosis . we describe the use of epidural balloon catheters to decompress the intervertebral foramen in three patients with spinal stenosis . patients were followed - up for 24 weeks . one patient reported moderate pain relief , three patients reported symptom improvement and one patient reported no change in symptoms . this report suggests that transforaminal balloon decompression using a balloon may have potential in the nonsurgical treatment of spinal stenosis by modifying the underlying pathophysiology of segmental spinal stenosis . |
| we describe a 26-year - old woman who presented with a nodular rash on the elbows following an insect bite . two days later , she developed erythema nodosum . both these lesions were treated symptomatically . one week later , she had purpura , abdominal pain , hematuria , and arthralgias , following which steroids were administered . her investigations revealed only microscopic hematuria that disappeared with therapy . this pattern of sequential appearance of rash and a nodular morphology are both unique features not previously reported . | we report a case of herpes simplex purpura ( hsp ) that presented with a sequential pattern of rashes following an insect bite . the patient was a 26-year - old woman who presented to our outpatient department ( opd ) with a nodular rash on her elbows and erythema nodosum on her lower limbs following an insect bite . she had purpura on her lower limbs 2 weeks later . she had similar lesions on both upper and lower limbs 1 week after a second insect bite . this pattern of rashes has not been previously reported in hsp . |

Table 19: Comparison of ground true abstract and abstract generated by REFRAG for PubMed.

| Ground True Abstract | Generated Abstract |
|---|---|
| memory - based learning , keeping full memory of learning material , appears a viable approach to learning nlp tasks , and is often superior in generalisation accuracy to eager learning approaches that abstract from learning material . here we investigate three partial memory - based learning approaches which remove from memory specific task instance types estimated to be exceptional . the three approaches each implement one heuristic function for estimating exceptionality of instance types : ( i ) typicality , ( ii ) class prediction strength , and ( iii ) friendly - neighbourhood size . experiments are performed with the memory - based learning algorithm ib1-ig trained on english word pronunciation . we find that removing instance types with low prediction strength ( ii ) is the only tested method which does not seriously harm generalisation accuracy . we conclude that keeping full memory of types rather than tokens , and excluding minority ambiguities appear to be the only performance - preserving optimisations of memory - based learning . | memory - based learning is a prominent and successful approach to machine learning . however , its use in nlp tasks has been limited by its over - generalisation properties . this paper investigates two optimisations that can be applied to memory - based learning in order to improve its generalisation performance : ( 1 ) replacing instance types ( memory tokens ) by instance types with frequency information , and ( 2 ) removing redundant information ( i.e. , instance types with low predictive power ) . we perform experiments on a large data set of english word pronunciations . we show that both optimisations yield improvements in generalisation performance . our results also indicate that atypicality , non - typicality , and friendly - neighbourhood size are estimates of the importance of instance types for their classification , rather than their removability . |
| machine learning methods are applied to finding the green s function of the anderson impurity model , a basic model system of quantum many - body condensed - matter physics . different methods of parametrizing the green s function are investigated ; a representation in terms of legendre polynomials is found to be superior due to its limited number of coefficients and its applicability to state of the art methods of solution . the dependence of the errors on the size of the training set is determined . the results indicate that a machine learning approach to dynamical mean - field theory may be feasible . | ewe demonstrate how machine learning techniques can be used to solve the quantum many - body problem . we apply the method to the anderson impurity model , where we use the exact diagonalization method to train the machine learning algorithms . we show that machine learning is an efficient and accurate method to compute the self - energy of the model and to predict the spectral function of the model . we also show that machine learning algorithms can be used to efficiently compute the self - consistent green s function starting from any hybridization function . |
| particle swarm optimization is used in several combinatorial optimization problems . in this work , particle swarms are used to solve quadratic programming problems with quadratic constraints . the approach of particle swarms is an example for interior point methods in optimization as an iterative technique . this approach is novel and deals with classification problems without the use of a traditional classifier . our method determines the optimal hyperplane or classification boundary for a data set . in a binary classification problem , we constrain each class as a cluster , which is enclosed by an ellipsoid . the estimation of the optimal hyperplane between the two clusters is posed as a quadratically constrained quadratic problem . the optimization problem is solved in distributed format using modified particle swarms . our method has the advantage of using the direction towards optimal solution rather than searching the entire feasible region . our results on the iris , pima , wine , and thyroid datasets show that the proposed method works better than a neural network and the performance is close to that of svm . * keywords * quadratic programming ; particle swarms ; hyperplane ; quadratic constraints ; binary classification . | support vector machines are used for classification of data in machine learning . support vector machines use quadratic programming formulation for minimizing the objective function . the quadratic programming problem is solved by particle swarm optimization . the proposed method is compared with khachiya s and karman s support vector machine algorithms for linear and neural networks and quadratic programming . the results show that the proposed method is better than the other two methods . |

Table 20: Comparison of ground true abstract and abstract generated by REFRAG for ArXiv.

Table 21: Performance on summarization tasks under the same latency.

| | Arxiv | | | Pubmed | | |
|---|---|---|---|---|---|---|
| | Rouge-1 | Rouge-2 | Rouge-L | Rouge-1 | Rouge-2 | Rouge-L ↑ |
| # Decoder tokens = 128 | | | | | | |
| LLAMA$_{FT}$ | 29.69 | 6.89 | 18.28 | 29.79 | 8.37 | 18.41 |
| CEPED | 12.67 | 1.66 | 8.39 | 12.01 | 1.41 | 7.74 |
| REPLUG$_{FT}$ | 5.30 | 0.78 | 3.77 | 5.11 | 0.81 | 3.55 |
| REPLUG$_{Chat}$ | 15.11 | 1.58 | 9.80 | 14.94 | 1.51 | 9.40 |
| LLAMA-32K | 2.83 | 0.48 | 2.11 | 7.94 | 1.63 | 5.31 |
| REFRAG$_8$ | **36.50** | **12.48** | **22.21** | **38.27** | **13.91** | **23.20** |
| REFRAG$_{16}$ | **38.48** | **12.50** | **22.66** | **38.93** | **12.83** | **23.07** |
| # Decoder tokens =512 | | | | | | |
| LLAMA$_{FT}$ | 36.03 | 11.16 | 21.49 | 38.15 | 14.36 | 23.27 |
| CEPED | 19.28 | 3.16 | 12.22 | 17.60 | 2.43 | 10.89 |
| REPLUG$_{FT}$ | 28.33 | 6.42 | 17.04 | 28.29 | 7.59 | 16.97 |
| REPLUG$_{Chat}$ | 31.41 | 7.00 | 18.32 | 30.67 | 7.13 | 17.56 |
| LLAMA-32K | 3.03 | 0.65 | 2.28 | 8.49 | 2.54 | 5.47 |
| REFRAG$_8$ | **41.95** | **15.56** | **24.84** | **43.55** | **17.53** | **26.38** |
| # Decoder tokens =1024 | | | | | | |
| LLAMA$_{FT}$ | 41.24 | 15.07 | 24.45 | 42.45 | 17.58 | 26.11 |
| CEPED | 25.20 | 5.07 | 15.45 | 23.00 | 3.94 | 13.71 |
| REPLUG$_{FT}$ | 19.32 | 3.18 | 12.73 | 17.07 | 2.93 | 11.20 |
| REPLUG$_{Chat}$ | 27.38 | 5.46 | 16.84 | 27.89 | 5.16 | 15.93 |
| LLAMA-32K | 4.34 | 0.95 | 3.35 | 10.19 | 3.11 | 6.47 |
| REFRAG$_8$ | **43.88** | **17.03** | **26.01** | **44.43** | **18.06** | **26.85** |

