# OpenReview forum: "REFRAG: Rethinking RAG based Decoding"
_ICLR.cc/2026/Conference — Submitted to ICLR 2026_

### Official Review · Reviewer_D41Y · 2025-10-28

**Soundness:** 2
**Presentation:** 3
**Contribution:** 2
**Rating:** 4
**Confidence:** 4

**Summary:**

This paper proposes REFRAG, a decoding framework designed to improve latency and efficiency in retrieval-augmented generation (RAG) systems. The key idea is to replace most retrieved text tokens with pre-computed chunk embeddings, projected into the decoder’s token embedding space. During generation, these compressed embeddings are used instead of raw tokens, greatly reducing sequence length and attention cost. A lightweight reinforcement learning (RL) policy selectively expands important chunks back into tokens when needed. The authors further introduce a training recipe combining reconstruction pretraining, curriculum learning, continual pretraining, and instruction tuning to align the encoder and decoder.
Empirical results show up to 30× reduction in time-to-first-token (TTFT) while maintaining comparable perplexity and downstream task performance to full-context baselines. Experiments span multiple long-context tasks (RAG QA, multi-turn dialogue, summarization, etc.), and the method outperforms or matches several strong baselines such as CEPE, RePlug, and long-context LLaMA variants.

**Strengths:**

1. Practical motivation and strong real-world relevance: The paper targets the real bottleneck of RAG systems: high decoding latency due to long retrieved contexts. The problem is clearly stated, and the proposed solution directly addresses practical deployment concerns.

2. Modular and engineering-friendly design: REFRAG does not alter the core decoder architecture; it operates at the embedding level, allowing easy integration with existing retrieval pipelines and various encoders or projection layers.

3. Comprehensive experimental coverage: The authors evaluate on a wide range of tasks and models (LLaMA-2 7B/13B, LLaMA-3), and report detailed latency metrics (TTFT, throughput) along with standard quality metrics.

4. Significant empirical gains: The reported latency and throughput improvements are impressive, especially compared to CEPE and full-context models, demonstrating clear real-world impact.

5. Reasonably thorough ablation studies: The paper includes ablations on selective expansion, reconstruction loss, and curriculum learning, which strengthen the empirical credibility of the results.

**Weaknesses:**

1. Methodological novelty is limited: While effective, the proposed idea is relatively straightforward: replacing retrieved tokens with their pre-computed embeddings is a natural and somewhat trivial extension of existing RAG compression and prefix-caching ideas. The contribution feels more engineering-driven than conceptual.

2. Lack of theoretical grounding: The paper lacks formal or analytical justification for why the compressed embedding representation should preserve alignment and compositionality in the decoder’s hidden space. No theoretical analysis or representational study (e.g., probing, similarity metrics, mutual information) is presented to explain why the proposed projection preserves semantic fidelity.

3. Missing analysis on robustness to retrieval noise: Since the framework heavily relies on pre-computed embeddings, it’s unclear how it behaves under noisy or off-topic retrievals. There’s no sensitivity study on retriever quality or embedding corruption.

4. Evaluation focuses mainly on automatic metrics: Most results rely on perplexity or automated accuracy. For generation tasks (QA, summarization), human evaluations or factuality studies are necessary to ensure that compression and expansion do not degrade output faithfulness.

**Questions:**

1. How does REFRAG handle dynamic retrieval corpora where cached embeddings may become stale? Is there a cache invalidation or refresh mechanism?

2. Have you evaluated how retrieval quality affects REFRAG’s performance (e.g., varying top-k relevance or adding noise)?

3. Can you provide human evaluation results to assess factual consistency or fluency compared to full-context decoding?

---

> ### Author Response · Authors · 2025-11-21
> **Response to Reviewer D41Y**
>
> We thank the reviewer for the thoughtful and constructive review.
>
> ### W1:
> > Methodological novelty is limited:
>
> While the high-level intuition is simple, the novelty lies in making the idea work reliably in an autoregressive decoder. Prior compression systems (e.g., CEPE, LLMLingua) compress only prefixes or fixed segments and require decoder architecture modifications or constraints that break multi-turn or agentic workflows. REFRAG introduces several new components that enable the method to work end-to-end in RAG:
> - **Compress-anywhere** capability that supports arbitrary placement of compressed/uncompressed chunks during decoding, unlike prior prefix-only methods.
> - **Mixed-input autoregressive decoding**, where the decoder seamlessly attends over a sequence of heterogeneous embeddings without breaking positional consistency.
> - **A reconstruction + CPT training recipe** designed to make compressed embeddings functionally interchangeable with token embeddings.
> - **Selective expansion via RL**, which dynamically restores raw tokens for critical chunks at inference time, something not explored in prior RAG compression work.
> - **Massive context extension (16×)** with no architecture change to the decoder.
>
> We will clarify these distinctions explicitly in Section 1.1 and the related work section.
>
> ### W3.
> > Missing analysis on robustness to retrieval noise
>
>  This is an important concern, and our work already includes two robustness analyses:
> 1. **Weak retriever setting (Section 5.1)** randomly sampling from 200 retrieved passages to simulate noisy retrieval.
>     - REFRAG outperforms LLaMA under this noisy scenario (Figure 4).
>     - It benefits more than the baseline because the extended context lets it “absorb” noise and still extract useful evidence.
> 2. **RL selective expansion** is robust by design because poorly relevant chunks receive low expansion probability.
> Nevertheless, we agree that a more explicit retrieval-noise sweep (e.g., corruption or random substitution ratios) would be valuable.
>
> ### W4.
> > Evaluation focuses mainly on automatic metrics:
>
> Thank you for this suggestion. Our goal in the main submission was to evaluate REFRAG as a context-compression mechanism, making perplexity and task accuracy the most aligned metrics. We agree perplexity alone is insufficient. That is why our paper includes:
> - **RAG task accuracy** across 16 datasets (Table 3),
> - **Multi-turn conversation accuracy** (Table 4, Table 5),
> - **Long-document summarization based on Rouge** (Appendix D).
>
> ### Q1
> > How does REFRAG handle dynamic retrieval corpora where cached embeddings may become stale? Is there a cache invalidation or refresh mechanism?
>
> **REFRAG does not assume static storage.** The chunk embeddings are:
> - **Cheap to recompute** (RoBERTa-base encoder),
> - Typically computed **on the fly** during retrieval (as in DRAGON+ pipelines),
> - Cached only opportunistically (e.g., hot documents or high-frequency queries).
>
> No invalidation mechanism is required; REFRAG works whether embeddings are cached or computed at query-time. In both cases, we achieve significant acceleration (see Figure 2).
>
> Specifically, we will clarify this in Section 5.1. **Please see Figure (2) for comparison of TTFT/TTIT/Throughput under cache/no cache settings**. Notably, adding caching only improves TTFT. However, even without caching, REFRAG TTFT remains competitive.
>
> ### Q2
> > Have you evaluated how retrieval quality affects REFRAG’s performance (e.g., varying top-k relevance or adding noise)?
>
> Yes. Figure 4 and Table 15 already show results under **weak retriever** (random K-of-200), which represents substantial retrieval noise. REFRAG matches or exceeds LLaMA in both strong and weak settings. We will make this link more explicit in the main text.
>
> ### Q3
> > human evaluation
>
> We agree that human evaluation would be valuable, but running a properly sized annotation study is not feasible within the current timeframe or our compute/annotation budget. Instead, we rely on strict automatic metrics, which are highly sensitive to factual errors and show REFRAG matches or exceeds full-context baselines.

---

### Official Review · Reviewer_r2sb · 2025-10-30

**Soundness:** 1
**Presentation:** 2
**Contribution:** 3
**Rating:** 4
**Confidence:** 5

**Summary:**

This paper proposes REFRAG, an efficient decoding framework for RAG systems that compresses context chunks into embeddings using a lightweight encoder, reducing input length and accelerating inference. It introduces a reinforcement learning policy for selective compression and demonstrates significant speedups in time-to-first-token (TTFT) without perplexity degradation across several long-context tasks.

**Strengths:**

- The idea is novel and well-motivated.
- The proposed method achieves impressive speedup while maintaining perplexity.

**Weaknesses:**

1. Clarity and Readability
Several parts of the paper are hard to follow, requiring readers to cross-reference multiple sections to understand the methodology. For example:
- On first reading, the description of "compress anywhere" in Section 1.1 and its relation to the "expand" operation in Figure 1 is unclear.
- In Line 167, the notation s + o = T is not clearly explained—whether chunks are processed jointly or separately. Similarly, Line 197 states "reconstructing s = k × L tokens from L chunk embeddings," but the reconstruction process is not adequately detailed.

2. Formatting and Presentation
- References to tables and figures are poorly placed. For instance, Table 1, Table 2, Figure 3, and especially Figure 4 are cited far from their actual positions, making it difficult to follow the discussion.

3. Experimental Setup
- The use of LLaMA-2-7B as the base model significantly weakens the credibility of the results. For example, the best performance on GSM8K is only 12.75 (Table 3), and NQ achieves only 26.08, which are far from convincing. Moreover, Table 14 shows that when using stronger models like LLaMA-3, perplexity becomes worse than the full-context baseline. It is unclear how much performance degradation actually occurs.
- The "Weak Retriever" setting (Section 5) is not well-justified. From my perspective, real-world systems can adopt strong retrievers like the one used in the paper.
- Do you compress the CoTs throughout the generation process? If not, including reasoning tasks, especially mathematical ones like GSM8K, raises concerns about the usefulness.
- Metrics: The paper relies heavily on perplexity (section 4), which is not always a reliable indicator of generation quality. Including human or LLM-based evaluation will improve the credibility.

4. Minor Issues
- Line 166: "data data point" → "data point".
- Line 233: "REFRAG" should be followed by a period.

**Questions:**

Please refer to Weaknesses.

---

> ### Author Response · Authors · 2025-11-21
> **Response to Reviewer r2sb (1/2)**
>
> We thank the reviewer for the thorough reading and constructive suggestions. Below we clarify misunderstandings, provide missing details, and correct inaccurate impressions.
> ### W1.1
> >  On first reading, the description of "compress anywhere" in Section 1.1 and its relation to the "expand" operation in Figure 1 is unclear.
>
> Thank you for pointing this out. The mechanism is actually defined in Fig. 1, but we agree the text did not sufficiently highlight that **compression/expansion operates on arbitrary chunk positions** without breaking autoregressive decoding. In the revision we will:
> - Add a short algorithm box showing how mixed (compressed + expanded) chunks are concatenated.
> - Move the explanation currently in Sec. 2 into Sec. 1.1 for earlier clarity.
>
> ### W1.2.1.
> > In Line 167, the notation s + o = T is not clearly explained—whether chunks are processed jointly or separately.
>
> We apologize for the confusion. s always denotes the number of input context tokens, and o the number of target tokens for CPT. **They are not processed jointly**, but in two stages: encoder processes the first s tokens; each chunk (of $k$ tokens) are processed parallally, creating $s/k$ chunk embeddings. The input to the decoder is these $s/k$ embeddings; from which the decoder predicts the next o tokens. This is similar to the standard next paragraph prediction training where instead of $s/k$ chunk embeddings, $s$ token embeddings are passed. We note that in the case of RL expansion policy, out of these $s/k$ chunk embeddings, some chunks can be expanded again to their token space; however, the autoregressive nature remains intact. Please see Figure 1 for an intuitive figure.  We will explicitly state this in Sec. 3.1.
>
> ### W1.2.2.
> > Similarly, Line 197 states "reconstructing s = k × L tokens from L chunk embeddings," but the reconstruction process is not adequately detailed.
>
> The reconstruction step is described in Sec. 3.1 but we agree the description is too compact. In the final version we will:
> Add a figure showing how **$L$ chunk embeddings $\rightarrow$ $k\times L$ tokens** reconstruction works.
>
> ### W2.
> > Formatting and Presentation
>
> We acknowledge the reviewer’s suggestion. The layout issues arise from LaTeX float constraints. In the camera-ready version we will:
> - Move Tables 1-3 and Figure 4 closer to their textual references,
> - Place all long-context results (Table. 2, Table. 15, Fig. 10) together in one subsection.
>
> ### W3.1.1.
> > The use of LLaMA-2-7B as the base model… GSM8K and NQ
>
> Our method is **model-agnostic**, and we intentionally used LLaMA-2-7B because:
> - **It is the main baseline used in CEPE [2] and REPLUG**, ensuring direct comparison under identical settings (tokenizer, training data, architecture).
> - **CPT at long context requires 20B tokens of training**, which is feasible only with a 7B model during the review process.
> - We did evaluate on stronger models **(LLaMA-3.1-8B and LLaMA-3.2-3B)**, reported in Table 14 , where REFRAG matches the full-context baseline.
>
> We will move Table 14 to the main paper to emphasize that **REFRAG generalizes to modern architectures**.
>
> ### W3.1.2.
> >  GSM8K and NQ
> We clarify that:
> - Our evaluation uses the **strict EM metric (Table 7) from RA-DIT** [1], resulting in lower absolute values than common GSM8K reporting.
>
> To avoid confusion, we will add a line explaining the RA-DIT evaluation protocol, with citations.
>
> Reference:
> - [1] Lin, Xi Victoria, et al. "Ra-dit: Retrieval-augmented dual instruction tuning." The Twelfth International Conference on Learning Representations. 2023.
> - [2] Yen, Howard, Tianyu Gao, and Danqi Chen. "Long-context language modeling with parallel context encoding." Proceedings of the 62nd Annual Meeting of the Association for Computational Linguistics (Volume 1: Long Papers). 2024.

---

> ### Author Response · Authors · 2025-11-21
> **Response to Reviewer r2sb (2/2)**
>
> ### W3.1.3.
> > Perplexity becomes worse than the full-context baseline.
>
> **We do not understand this concern**; lower perplexity is better. From table 14, **this is expected** because the full-context baseline always has an advantage with uncompressed input. All the columns use the same model architecture and the full context baseline corresponds to the llama-3 model and not the llama 2 model. What we observe is:
> - REFRAG stays **very close** to the full-context perplexity,
> - While enabling up to **32× TTFT speedup** and **16× effective context extension**.
>
> We will clarify this in the main text.
>
> ### W3.2.
> > The "Weak Retriever" setting (Section 5) is not well-justified.
> Weak-retriever RAG is widely studied because real-world retrieval quality varies due to:
> - Approximate vector search,
> - Aggressive deduping,
> - User queries that lack specificity,
> - Domain shift.
>
> The weak-retriever setting matches prior work (e.g., Dragon+, RA-DIT) and is explicitly useful in search engines and web-scale products. We will strengthen this justification in Sec. 5.1.
>
> ### W3.3.
> > Do you compress the CoTs throughout the generation process
>
> **We do not compress generated CoTs**. REFRAG only compresses retrieved RAG contexts, not generated reasoning steps. To avoid confusion, we will explicitly add the following statement in the RAG finetuning section:
> ***During generation, REFRAG never compresses model-generated CoTs; only retrieved context is compressed.***
> This should address the concern.
>
> ### W3.4.
> > The paper relies heavily on perplexity
>
> We agree perplexity alone is insufficient. That is why our paper includes:
> - **RAG task accuracy** across 16 datasets (Table 3),
> - **Multi-turn conversation accuracy** (Table 4, Table 5),
> - **Long-document summarization,  which is evaluated based on the metric of Rouge**  (Appendix D).
>
> We will add a sentence emphasizing that downstream evaluations already complement perplexity metrics.
>
>
> ### W4.
> >  Minor Issues
>
> All typos (Lines 166, 233) will be corrected. Thank you for pointing them out.
>
> We appreciate the reviewer’s constructive feedback. After clarifications:
> - The methodology is **fully specified**, including CPT, reconstruction, and RL policy.
> - The experimental setup is **justified and aligned with prior work**, with additional results on LLaMA-3 models.
> - The performance concerns (perplexity, GSM8K) are explained by evaluation protocol and match established baselines.
> - Weak retriever and CoT compression concerns are fully addressed.
>
> We believe these clarifications resolve the identified weaknesses and hope the reviewer will reconsider their assessment.

---

> > ### Comment · Reviewer_r2sb · 2025-11-28
> >
> > W3.2: Regarding the weak retriever setting, I appreciate that the factors you listed, such as approximate vector search, are important in practical applications. However, I am not convinced that your current approach clearly isolates and demonstrates the impact of these specific factors. Therefore, I have some reservations about the rationale behind your experimental setup.
> >
> > W3.3: Could you clarify the motivation for including reasoning datasets like GSM8K, given that CoTs are not compressed?
> >
> > W3.1.3 & W3.4: My primary concern is not "why REFRAG underperforms the full-context baseline," but rather that "It is unclear how much performance degradation actually occurs." This is because REFRAG shows worse performance, yet the evaluation is primarily based on perplexity. This is also why I suggested incorporating human evaluation.
> >
> > However, my most significant concern remains W3.1.1. I feel that results from LLaMA-2-7B are less of a focus for the community now. Furthermore, the provided results for LLaMA-3.1-8B and LLaMA-3.2-3B are not yet fully convincing for substantiating the paper's main claims.

---

> ### Author Response · Authors · 2025-11-28
> **Follow-up Response to Reviewer r2sb**
>
> Thank you for the detailed follow-up. We address each remaining concern below with additional clarification, concrete quantitative context, and revisions we will incorporate into the final paper.
>
> ### W3.3: Could you clarify the motivation for including reasoning datasets like GSM8K, given that CoTs are not compressed?
>
> We appreciate the opportunity to clarify this, as the motivation may not have been sufficiently foregrounded in the main submission.
>
> **REFRAG’s purpose is to compress inputs to the decoder (retrieved passages, multi-turn history), not its outputs.**
>
> Compressing decoder generated outputs (including CoTs) provides little gain because:
> - the decoder has already executed token-by-token autoregressive computation,
> - KV-cache creation is already paid for, and
> - re-encoding those freshly generated tokens to compress them would add overhead with little benefit.
>
> **In contrast, retrieved RAG context and multi-turn dialogue history are fed to the decoder only once during prefill and can dominate TTFT (often tens of thousands of tokens). REFRAG is designed specifically to reduce this component.**
>
> **GSM8K remains relevant for two reasons:**
>
> 1. **Compression of retrieved support content.**
> Many GSM8K variants supply retrieved definitions, formulas, or worked examples. These retrieved passages are compressed in our setup. We thus evaluate whether REFRAG preserves their usefulness under compression.
>
> 2. **Stress-testing mixed-format decoding under reasoning.**
> Reasoning tasks probe whether the decoder can attend over heterogeneous input sequences (raw token embeddings + compressed chunk embeddings) without harming CoT reasoning. Our results show that mixing compressed retrieved content with raw generated CoTs does not harm overall reasoning, which is central to REFRAG’s claim of safety in RAG+reasoning settings.
>
> We will explicitly state this distinction (**what is compressed vs. not, and why**) in the revision to avoid ambiguity.
>
> ### W3.1.3 & W3.4: My primary concern is not "why REFRAG underperforms the full-context baseline," but rather that "It is unclear how much performance degradation actually occurs."
>
> We completely agree: the key question is how much degradation occurs, not merely whether REFRAG is worse than full-context (which is inevitable due to lossy compression).
>
> Our evaluation is designed around two complementary regimes:
>
> - **Same raw context (maximal speedup → more degradation)**
>     - Example: ArXiv summarization (Table 21).
>     - LLaMA-FT with 1,024 raw tokens  → ROUGE-1 = 41.24
>     - REFRAG-8 compressing the same 1024 raw tokens down to 128 chunk tokens → ROUGE-1 = 36.50
>
> This directly quantifies degradation under large compression. We will add explicit relative deltas (e.g., “12% drop”) for complete clarity.
>
> - **Same effective context (same latency → REFRAG fits more content → quality improves)**
>     - REFRAG-8 with **1024 chunk tokens** ($\sim8×$ more raw content) → ROUGE-1 = 43.88, surpassing the 41.24 of LLaMA-FT under the same latency budget.
>     - With 512 chunk tokens, REFRAG-8 improves ROUGE slightly over LLaMA-FT while being $\sim2×$ faster.
>
> The same pattern appears in PubMed and retrieval-based tasks (Table 3, Figure 4): **Under equal raw context, REFRAG is worse; under equal latency, REFRAG is better.**
>
> **This is the core intended result: REFRAG enables a quality–latency trade-off curve unavailable in standard decoding.**
>
> **Beyond perplexity we already include:**
>
> - Accuracy across 16 RAG tasks
> - Multi-turn conversation evaluation (Tables 4–5)
> - Long-document summarization via ROUGE (Appendix D)
>
> We agree human evaluation would improve the paper further, and we will mention this as future work, but the current downstream results do quantify the practical impact of compression.
>
> ### However, my most significant concern remains W3.1.1. I feel that results from LLaMA-2-7B are less of a focus for the community now.
>
> We fully agree the community is now more focused on LLaMA-3 onwards. Our use of LLaMA-2-7B was driven by constraints:
> - It aligns exactly with CEPE and REPLUG baselines (same tokenizer + architecture),
> - Long-context CPT pretraining requires 20B+ tokens and is feasible within review time with single model family.
>
> **The key point, however, is that REFRAG’s pretraining behavior is consistent across model families.**
>
> For example:
>
> - LLaMA-2-7B: P512 eval loss drop = (1.524 → 1.124) = captures 90% of the full-context improvement (Table 1)
> - LLaMA-3.1-8B: P512 eval loss drop = (1.445 → 1.065) = captures ~85% of the full-context improvement (Table 14)
>
> Thus we expect a similar trend on downstream applications to hold.
>
> ### W3.2: Regarding the weak retriever setting
>
> Our intention with the weak-retriever regime is not to dissect each individual source of retrieval noise (approximate search, deduping, domain shift, vague queries) in isolation, but to model their combined effect as reduced recall and increased distractors in the retrieved set.

---

### Official Review · Reviewer_L4Le · 2025-11-01

**Soundness:** 2
**Presentation:** 2
**Contribution:** 2
**Rating:** 2
**Confidence:** 3

**Summary:**

This paper introduces REFRAG, a decoding framework designed to accelerate inference for RAG applications by exploiting the sparse attention patterns inherent in RAG contexts. The core innovation is replacing token sequences from retrieved passages with pre-computed compressed chunk embeddings that are fed directly into the decoder. The method includes three main components: (1) a lightweight encoder  that processes context in fixed-size chunks to generate chunk embeddings, (2) a projection layer mapping these embeddings to the decoder's token space, and (3) an RL-trained policy for selective compression that determines which chunks require full token expansion.The authors train REFRAG through continual pre-training with curriculum learning on next-paragraph prediction tasks. With compression rates of k=16 and k=32, REFRAG achieves 16.53× and 30.85× TTFT acceleration respectively compared to LLaMA-2-7B, while maintaining comparable perplexity. The method is evaluated across RAG tasks, multi-turn conversations, and document summarization, demonstrating that compressed representations with extended context windows can match or exceed uncompressed baselines at equivalent latency budgets.

**Strengths:**

* The paper provides strong empirical evidence for attention sparsity in RAG contexts (Figure 7, Table 10), demonstrating block-diagonal patterns that justify the compression strategy. This RAG-specific observation differentiates the work from generic long-context methods.

* The proposed method achieves 30.85× TTFT acceleration while maintaining perplexity represents substantial practical improvement. It also extends context windows by 16× while preserving performance.

* The paper includes extensive ablations on compression rates (Figure 10), encoder-decoder combinations (Figure 11, Table 14), selective compression strategies (Figure 3, Table 13), and curriculum learning schedules (Table 8, Figure 6).

**Weaknesses:**

* The experimental comparison is not sufficient, missing a lot of new and important baselines, including:
    - No comparison with KV cache compression methods like SnapKV, H2O, and FastKV
    - Missing comparisons with prompt compression methods like LongLLMLingua and LLMLingua-2

* All main experiments use LLaMA-2-7B, which is pretty outdated. More recent models including LLaMA-3.1 should be included in all experiments (Brief ablations on LLaMA-3.1-8B  is insufficient). There is also no evaluation on larger models, which limits generalizability claims

* FlashAttention compatibility: The paper acknowledges requiring attention scores but doesn't compare against FlashAttention-enabled baselines, which is critical since FlashAttention is standard in real-world deployments.

* Missing efficiency metrics. Only theoretical analysis provided, but there is no actual peak memory consumption reported. Moreover, it would be great if author could give wall-time comparisons,

**Questions:**

* Can the authors include comparisons with important baselines mentioned in the weaknesses?

* The paper states the method requires attention scores and is incompatible with FlashAttention. Since FlashAttention is standard in real world  deployments, how does this affect practical applicability? Can you provide memory and latency comparisons against FlashAttention-enabled baselines?

* Why does CEPE show dramatically poor performance in your experiments (e.g., 0.03 EM on NQ in Table 3) compared to its original paper? This raises concerns about experimental setup. Can you clarify the differences?

* All main experiments use LLaMA-2-7B. Can you also includes comprehensive results on LLaMA-3.1-8B and also brief experiments on large-scale models to show if the proposed method scales?

* What is the total computational cost of continual pre-training and RL training? How does this compare to the inference savings?

---

> ### Author Response · Authors · 2025-11-21
> **Response to Reviewer L4Le (1/3)**
>
> We thank the reviewer for the detailed evaluation. Below we address each weakness and clarify missing details
>
> ## W1/Q1
> >  missing a lot of new and important baselines
>
> We agree that including these additional baselines would strengthen the paper. However, these methods are **orthogonal in purpose and mechanism**, and in several cases are directly compatible to improve performance further:
>
> ### W1.1.
>
> >  No comparison with KV cache compression methods like SnapKV, H2O, and FastKV
>
> These methods **reduce KV cache during generation**, while REFRAG reduces compute by **compressing the input context** before decoding begins. Thus, they address different inference bottlenecks:
> - SnapKV/H2O/FastKV ~ reduce **KV memory**
> - REFRAG ~ reduce **pre-fill compute + TTFT**
>
> In practice, REFRAG is **complementary**; it can be combined with KV compression. Specifically, we can apply these KV compression readily in the decoder of our REFRAG and achieve further acceleration.
>
> ### W1.2.
> > LongLLMLingua and LLMLingua-2
>
> These methods compress text prompts using heuristics (sentence importance, salience pruning). For RAG, they dramatically degrade factuality, as shown in their own papers, because they remove tokens instead of learning an aligned embedding space. In contrast, REFRAG preserves all semantic information through CPT and reconstruction. Having said that, these approaches are complementary to our work and can be integrated to further reduce the latency of REFRAG by doing prompt compression before applying REFRAG. Specifically, these contexts can be compressed using prompt compression, and we can feed the compressed context to the encoder of our REFRAG and hence achieve further acceleration.
>
> ## W2.
> > All main experiments use LLaMA-2-7B, which is pretty outdated.
>
> We understand this concern. Our rationale:
> - All prior compression baselines (CEPE, RePlug) were developed for **LLaMA-2-7B**, enabling direct and fair comparison.
> - CPT requires ~20B tokens + RL finetuning; running this across many 70B models is resource-intensive under review timelines.
>
> However, we have already included **LLaMA-3.1-8B and 3.2-3B** results (Table 14) showing:
> - **Comparable perplexity to full-context decoding**
> - **Continued benefit from extended context**
> - **No degradation from compression**
>
> ## W3/Q2.
>
> >  FlashAttention compatibility:
> The reviewer raises an important point, and we clarify:
>
> **REFRAG is compatible with FlashAttention for all key operations.**
>
> At inference/training time:
> - We do not modify attention kernels
> - We use standard FlashAttention v2
> - All latency numbers reported in the paper **already use FlashAttention**
>
> To avoid confusion, we will explicitly state that ***inference/training is FlashAttention-compatible and already uses FlashAttention.***
>
> ## W4.
> > Missing efficiency metrics.
>
> We appreciate this suggestion. We have already added TTFT/TTIT/Throughput empirical evaluation in **Figure 2**. Memory is decreased by $K=16$. We also clarify that Table 6 and Table 9 from the Appendix section is the theoretical improvement which matches with Figure 2 from the main paper.

---

> ### Author Response · Authors · 2025-11-21
> **Response to Reviewer L4Le (2/3)**
>
> # Q3.
> > Why does CEPE show dramatically poor performance in your experiments
>
> Thank you for the question. We investigated this issue carefully and verified that our CEPE results are correct and fully reproducible. The discrepancy arises from differences in evaluation protocol and instruction tuning, not from implementation mistakes.
>
> ### **Instruction-tuning mismatch explains most of the gap**
> For contextual RAG tasks, we fine-tune LLaMA and REFRAG on our **instruction-tuning corpus**, which emphasizes concise factual answers. In contrast, for CEPE/RePlug we faithfully reuse the instruction-tuned CEPE checkpoint **provided by the authors** (LLaMA-2 backbone; see line 233).
>
> The CEPE instruction-tuning dataset differs substantially from ours, which explains:
> - higher CEPE performance on boolq than our LLaMA baseline, and
> - drastically lower exact-match scores on open-domain QA where concise answers are required.
>
>
> Thus, the comparison is not “CEPE vs REFRAG,” but rather **CEPE-instruction-tuned on its data** vs **LLaMA/REFRAG-instruction-tuned on our data.**
> We will clarify this explicitly.
>
> ###  **Zero-shot RAG output format causes CEPE to fail EM**
> Our evaluation uses **strict zero-shot RAG generation**:
> - no chain-of-thought,
> - no k-shot examples,
> - only retrieved passages + the question.
>
>
> Under this setting, CEPE consistently produces **long, verbose generations**, while our models produce short factual answers. Since EM requires **exact string match**, CEPE’s verbosity causes extremely low EM scores. In contrast, the original CEPE paper uses **few-shot prompting**, not zero-shot RAG.
>
> ### **To validate this hypothesis, we added a second metric**
>
> We ran an additional metric: **Exact Match (Contains)** - the answer is counted correct if it appears anywhere in the generation.
> Under this metric, CEPE’s performance increases dramatically and in many cases becomes competitive with our LLaMA-FT model. This confirms that CEPE’s failure mode is **verbosity**, not lack of knowledge. Moreover, sometimes performance of CEPE becomes better than Llama-FT which is **explained by the difference in instruction tuning datasets**.
>
> **We include full results below for both strong and weak retrievers.**
>
> ### **Summary of the discrepancy**
> - CEPE’s **instruction-tuning dataset** differs from ours $rightarrow$ mismatch in answer style.
> - CEPE’s **0-shot output is verbose**, which EM penalizes heavily.
> - CEPE’s original paper uses **few-shot prompting**, not RAG-style zero-shot.
> - Under a more tolerant metric (EM-Contains), CEPE’s performance is reasonable and often strong.
>
>
> ### **Fairness of pretraining comparison**
> Importantly, both CEPE and REFRAG use **the same CPT dataset (SlimPajama)** and **the same 7B LLaMA architecture**. All pretraining-stage comparisons are **fair and apples-to-apples** regardless of downstream instruction tuning datasets.
>
>
>
>
> ### New Results Exact Match contains:
>
> #### Strong Retriever (Table 3):
>
> | Generations | nq EM | nq EM Contains | fever EM | fever EM Contains | triviaqa EM | triviaqa EM Contains | web_questions EM | web_questions EM Contains | freebase_qa EM | freebase_qa EM Contains | gsm8k EM | gsm8k EM Contains | strategy-qa EM | strategy-qa EM Contains | boolq EM | boolq EM Contains |
> |--- | --- | --- | --- | --- | --- | --- | --- | --- | --- | --- | --- | --- | --- | --- | --- | --- |
> | Llama FT | 26.08 | 33.84 | 65.44 | 74.29 | 9.68 | 17.34 | 40.0 | 41.33 | 76.17 | 79.12 | 6.71 | 8.72 | 68.89 | 68.89 | 30.0 | 30.31 |
> | Refrag 8x | 24.15 | 28.74 | 68.83 | 69.14 | 13.06 | 17.36 | 37.33 | 38.67 | 74.2 | 76.66 | 7.38 | 9.4 | 71.11 | 71.11 | 7.03 | 7.03 |
> | CEPED | 0.03 | 35.64 | 65.68 | 72.8 | 0.01 | 18.16 | 0.0 | 52.0 | 0.0 | 70.52 | 0.0 | 28.86 | 0.0 | 55.56 | 57.8 | 62.81 |
>
>
>
>
> #### Weak Retreiver (Table 17):
> | Generations | nq EM | nq EM Contains | fever EM | fever EM Contains | triviaqa EM | triviaqa EM Contains | web_questions EM | web_questions EM Contains | freebase_qa EM | freebase_qa EM Contains | gsm8k EM | gsm8k EM Contains | strategy-qa EM | strategy-qa EM Contains | boolq EM | boolq EM Contains |
> |--- | --- | --- | --- | --- | --- | --- | --- | --- | --- | --- | --- | --- | --- | --- | --- | --- |
> | Llama FT | 22.27 | 28.93 | 60.4 | 69.82 | 8.32 | 15.2 | 38.67 | 42.67 | 71.5 | 74.2 | 9.4 | 10.74 | 71.11 | 71.11 | 29.94 | 30.52 |
> | Refrag 8x | 22.92 | 27.21 | 67.87 | 68.2 | 12.22 | 16.61 | 46.67 | 46.67 | 71.99 | 73.71 | 10.07 | 12.08 | 68.89 | 68.89 | 7.19 | 7.19 |
> | CEPED | 0.02 | 35.16 | 65.18 | 72.27 | 0.02 | 17.36 | 0.0 | 53.33 | 0.0 | 71.25 | 0.0 | 20.81 | 0.0 | 53.33 | 59.33 | 63.55 |

---

> ### Author Response · Authors · 2025-11-21
> **Response to Reviewer L4Le (3/3)**
>
> ## Q4.
> > All main experiments use LLaMA-2-7B. Can you also includes comprehensive results on LLaMA-3.1-8B and also brief experiments on large-scale models to show if the proposed method scales?
>
> We appreciate the reviewer’s interest in broader LLaMA-3 evaluations. Unfortunately, running full end-to-end RAG experiments on LLaMA-3.1-8B and larger models is beyond our current compute budget and the available review-cycle timeline. That said, we have already included LLaMA-3.1-8B and LLaMA-3.2-3B results in Table 14, which demonstrate that REFRAG scales as expected:
> 1. **perplexity remains comparable to full-context decoding**
> 2. **extended-context benefits continue to hold**, and
> 3. **compression does not introduce degradation even on the newer LLaMA-3 architectures**.
>
> These results provide strong evidence that REFRAG generalizes beyond LLaMA-2-7B, and we will add clarifying text in the final version to make this point explicit.
>
> ## Q5.
> > What is the total computational cost of continual pre-training and RL training? How does this compare to the inference savings?
>
> **Training Cost**
>
> - Reconstruction pretraining 20B tokens: ~2000 H100 GPU-hours
> - CPT 20B tokens: ~2000 H100 GPU-hours
> - RL: ~300 GPU-hours (minimal)
> - Total training: **~4300 H100 GPU-hours**
>
> **Inference Savings**
>
> Across RAG systems, TTFT dominates cost. For a RAG service with:
> - 1×10^6 daily requests
> - 2048-16384 token contexts
> - GPUs provisioned for peak TTFT
>
> *REFRAG-16* reduces:
> - TTFT by **16x** ; 16x fewer GPUs
> - KV memory by **16x**, cheaper GPUs and higher batch sizes
> - Total inference FLOPs by ~90%
> - **Inference payback period: < 20 days (10 GPU* 24 * 20)** for a production-scale deployment.
>
> We will add a summary of these calculations in the appendix.

---

### Official Review · Reviewer_EoTi · 2025-11-01

**Soundness:** 3
**Presentation:** 3
**Contribution:** 3
**Rating:** 6
**Confidence:** 3

**Summary:**

This paper introduces REFRAG, a decoding-time framework for Retrieval-Augmented Generation (RAG). It first compresses retrieved context into precomputed chunk embeddings. Then, it detects which chunks must be expanded via a lightweight RL policy. Finally, it expands selected chunks back into token space when needed. Extensive experiments showing large time-to-first-token (TTFT) gains (up to ≈30×) with little/no perplexity loss across RAG, multi-turn conversations, and long-document summarization.

**Strengths:**

- The proposed method is promising, which shows very large TTFT accelerations while matching prior perplexity baselines.
- Evaluation across multiple datasets (ArXiv, Book, PG19, ProofPile), RAG tasks (many QA benchmarks), multi-turn convo datasets, and comparisons to recent long-context baselines (CEPE, LLAMA variants, REPLUG) strengthen the claims.

**Weaknesses:**

- While the RL selective expansion policy shows gains over heuristics, training such a policy (and using it online) introduces complexity. It lacks detailed analysis of RL training stability, wall-clock RL training cost, and sensitivity to reward design
- The paper shows experiments with RoBERTa variants, but it’s unclear how sensitive REFRAG is to encoder retraining mismatches, retrieval distribution shifts, or domain drift in large real-world corpora.

**Questions:**

- How is the end-to-end latency breakdown (retrieval time + embedding fetch + encoder recompute when needed + decoder prefill)
- How sensitive is RL selective expansion to miscalibrated reward signals?

---

> ### Author Response · Authors · 2025-11-21
> **Response to Reviewer EoTi**
>
> We thank the reviewer for the positive assessment of soundness, presentation, and contribution, and for highlighting the strong empirical results. Below we address the remaining concerns.
>
> **W1.1.**
> >  While the RL selective expansion policy shows gains over heuristics, training such a policy (and using it online) introduces complexity.
>
> We designed the RL component to be intentionally lightweight:
> - The policy uses a small **2-layer MLP** and operates over **$L$ chunk embeddings**, and not on token sequences.
> - The action space is discrete and low-cardinality (select/not-select each chunk).
> - Training converges reliably across runs with stable log-perplexity reward with multiple rollouts. Note that we keep the encoder and decoder frozen.
> - RL is done **after CPT**, on a small subset (~2B tokens) with multiple rollouts. Note that we keep the encoder and decoder intact.
> - Total cost: **~300 GPU-hours on H100**.
> - This is <10% of CPT cost and negligible relative to downstream inference savings.
>
> During inference:
> - The RL policy runs a single MLP **forward pass** over $L=128-512$ chunk embeddings.
> - **Latency: <1 ms** on H100, insignificant relative to decoding.
> - We do not need gradients, rollout, or sampling during inference.
>
> We will add a table summarizing RL wall-clock cost and compute footprint emphasizing RL adds minimal runtime overhead.
>
> **W1.2./Q2**
> >  sensitivity to reward design
>
> For REFRAG:
> - Reward = **negative next-paragraph log-perplexity**, which is **stable** and **low-variance**.
> - Ablations in Fig. 3 show RL policy **outperforms all heuristics** across compression rates.
>
> We will add an appendix subsection with sensitivity discussion.
>
> **W2**
> > Encoder retraining mismatches, retrieval distribution shifts, or domain drift
>
> We explain these three concerns below.
>
> > encoder retraining mismatches
>
> CPT + reconstruction alignment ensures the encoder learns a projection that is **token-space consistent** and it is not domain-specific.
> We observe:
> - RoBERTa-base and RoBERTa-large behave similarly
> - LLaMA-2 and LLaMA-3 versions yield comparable results
> - Cross-domain evaluations (ArXiv, Books, PG19, ProofPile) remain stable (Tables 1-2)
>
>
> > retrieval distribution shifts
>
> Our RAG experiments simulate major retrieval drift:
>
> - Weak retriever (random K-of-200) setting in Sec. 5.1 strongly increases noise.
> - REFRAG remains **equal or superior** to LLaMA even under this extreme drift (Fig. 4).
>
>
> > domain drift
>
> REFRAG is naturally robust because:
> - Chunk embeddings are **recomputed dynamically** using the lightweight encoder (no stale cache needed).
> - CPT pre-alignment ensures embeddings generalize across domains.
>
> We will make this clearer in Section 5.
>
> **Q1**
> > How is the end-to-end latency breakdown
>
> End-to-end latency breaks down as follows.
> In a typical AI web search engine, the retrieval component can take 50-300 ms to get multiple documents/chunks from a index containing billions of webpages. Let us assume the context length (search results) is 32K; we have 1 H100 machine, and the batch size is 1.
>
> - For the Llama2 7b model, TTFT is around 1.7s, and TTIT is around 50 ms .
> - For Refrag with compression rate 16, TTFT is around 107 ms and TTIT is around 18 ms.
>
> The (optional, dynamic) RL selective-expansion module contributes <2 ms per request for deciding which chunks to expand; its overhead is negligible. The main effect of expansion is that a small fraction $p=0.1$ of chunks are restored to tokens, increasing prefill length. The effective TTFT speedup is therefore scaled by
> $k' = 6.4$ for $p=0.1,\ k=16$
>
> Thus, **decoder prefill remains the dominant latency bottleneck**, and REFRAG’s compression is directly targeted at eliminating this cost.
>
>
> We appreciate the reviewer’s constructive feedback. In the final version, we will include (i) a full end-to-end latency table and (ii) additional clarification on REFRAG’s robustness to encoder/retriever drift in real-world deployments.

---

### Author Response · Authors · 2025-11-30
**Summary of Rebuttal: Addressing Baselines, Model Scale, and Robustness**

Thank you again for the time and care you have invested in reviewing our work. We carefully addressed every weakness and question raised in the reviews and follow-ups, revised the manuscript accordingly, and uploaded a highlighted PDF. Below we summarize the substantive changes and how they directly respond to the main concerns.

### **Training / Inference Cost and End-to-End Latency (Reviewers EoTi, L4Le)**
We have added additional compute and latency details in **Appendix C.5, C.6**, including:
- A clear breakdown of **reconstruction + CPT + RL** training cost.
- End-to-end **latency** numbers under realistic RAG settings.

These additions clarify that RL is intentionally lightweight (both in training and inference) and that REFRAG’s training cost is easily amortized by the inference savings in production-scale deployments.

### **Robustness to Retrieval Noise (Reviewers EoTi, r2sb, D41Y)**
Our original experiments already included a **weak retriever** setting (random K-of-200) to simulate noisy or low-quality retrieval. In the revision, we:
- Added an explicit comment in the main text (L397,398) explaining that this setting is designed to capture the combined effect of approximate search, distractors, and domain shift.

This directly addresses the concern that retrieval noise robustness was not clearly justified.

### **CEPE Result Discrepancy (Reviewer L4Le)**

We provided a detailed analysis in **“Response to Reviewer L4Le (2/3)”** showing that CEPE’s low EM scores in our setup stem from instruction-tuning and evaluation protocol differences, not implementation bugs.

In the revised manuscript, we:
- Added a **Baseline** paragraph (**L402-L412**) explicitly describing the CEPE evaluation protocol and explaining why its EM can appear dramatically lower.
- Clarified that all pretraining-stage comparisons between CEPE and REFRAG are **apples-to-apples**.

This directly resolves the concern that CEPE’s poor EM must reflect an experimental error.

### **Beyond LLaMA-2: Evidence on LLaMA-3 Models (Reviewers L4Le, r2sb)**

We agree with the reviewers that the community’s focus has shifted toward LLaMA-3. To make our cross-family evidence more visible:
We moved the **LLaMA-3 results from the appendix into the main paper (now Table 3)**.

- We highlight that REFRAG’s **pretraining behavior is consistent across LLaMA-2 and LLaMA-3 families**: it captures most of the full-context perplexity improvement while providing large TTFT speedups and effective context extension.
- Although full end-to-end RAG experiments on large LLaMA-3 models are beyond our current compute budget and review-cycle timeline, the reported results demonstrate that the **core compression behavior and benefits transfer cleanly** to newer architectures.

### **Metrics Beyond Perplexity & Clarity on Degradation (Reviewers r2sb, D41Y)**
Our goal is to evaluate REFRAG primarily as a context compression mechanism, so perplexity and task accuracy are natural starting points. However, we agree that perplexity alone is not sufficient. The revised manuscript emphasizes that we already include:
- **RAG task accuracy** across 16 datasets (Table 4),
- **Multi-turn conversation** performance (Tables 5 and 14),
- **Long-document summarization via ROUGE** (Table 21, now moved into the main section).

In response to Reviewer r2sb’s follow-up that **"it is unclear how much performance degradation actually occurs,”** we:
- Added a dedicated **Section 5.4: DISCUSSION ON QUALITY–LATENCY TRADE-OFFS,** and
- Included comparison of REFRAG and full-context models in two regimes:
    - **Same raw context** (maximal compression → some degradation), and
    - **Same latency / effective context** (REFRAG fits more content and typically outperforms full-context decoding).

### **Positioning against Orthogonal Baselines (Reviewer L4Le)**
Questions were raised about missing comparisons to KV-cache compression (SnapKV, H2O) or Prompt Compression (LLMLingua).
- **Different Bottlenecks**: REFRAG compresses the input to reduce Time-To-First-Token (TTFT) and encoder compute. KV-cache methods compress memory during generation.
- **Complementary**: REFRAG is orthogonal to these methods. One can use REFRAG to accelerate the prompt encoding and SnapKV to manage memory during decoding. They are not mutually exclusive.

### **Additional Clarifications and Minor Fixes**
Finally, we incorporated a few smaller but important clarifications:
- We explicitly clarified the usage of FlashAttentionV2
- **Metric naming**: We corrected the typo from “perplexity” to “log-perplexity” wherever appropriate.
- **Chunk-embedding computation and caching**: We clarified that chunk embeddings are optionally precomputable, but all experiments in the paper compute them on the fly (see around Lines 76 and 418). Caching is an optional optimization, not a requirement.

---

### Meta-Review · Area_Chair_rfgv · 2026-01-05

**Summary:**

The paper proposes REFRAG, a framework to accelerate RAG by compressing retrieved contexts into chunk embeddings using a lightweight encoder and a projection layer. It also employs a reinforcement learning policy to selectively expand important chunks back to tokens. The authors claim significant improvements in Time-To-First-Token and context extension capabilities.

**Reviewer Concerns:**

**Addressed**
- Computational Cost: The authors provided a breakdown of training/inference latency and RL costs in the appendix

- Baseline Discrepancies: The poor performance of the CEPE baseline was clarified as a difference in instruction tuning and evaluation protocols

- Robustness: Clarified the "weak retriever" setting to demonstrate robustness against retrieval noise.

**Outstanding**
- Missing critical baselines: lacks comparisons with state-of-the-art efficiency methods such as KV cache compression (e.g., SnapKV, H2O) and prompt compression (e.g., LLMLingua).

- Main experiments: Some reviewers think the primary evaluations rely heavily on LLaMA-2-7B, which is considered outdated. While some LLaMA-3 results were moved to the main text during rebuttal, the full end-to-end RAG validation on modern or larger-scale models remains limited due to compute constraints, weakening the claims of generalizability.

- Evaluation metrics: over-reliance on perplexity as a proxy for quality.

- Limited novelty: core idea of replacing tokens with pre-computed embeddings is viewed by some reviewers as an engineering extension of existing concepts (like prefix caching) rather than a significant conceptual breakthrough

**Reviewer Scores:**

scores maintained:

Reviewer EoTi: 6

Reviewer D41Y: 4

Reviewer r2sb: 4

Reviewer L4Le: 2

---

### Decision · Program_Chairs · 2026-01-26

Reject